# SENDAI: A Hierarchical Sparse-measurement, EfficieNt Data AssImilation Framework

**Xingyue Zhang**[*]
School of Environmental and Forest Sciences
University of Washington
Seattle, WA 98195, USA
`xyzhamy@gmail.com`

**Yuxuan Bao**[*]
Department of Applied Mathematics
University of Washington
Seattle, WA 98195, USA
`baoyx@uw.edu`

**Mars Liyao Gao**
Paul G. Allen School of Computer Science & Engineering
University of Washington
Seattle, WA 98195, USA
`marsgao@uw.edu`

**J. Nathan Kutz**
School of Environmental and Forest Sciences
University of Washington
Seattle, WA 98195, USA
`kutz@uw.edu`

## ABSTRACT

Bridging the gap between data-rich training regimes and observation-sparse deployment conditions remains a central challenge in spatiotemporal field reconstruction, particularly when target domains exhibit distributional shifts, heterogeneous structure, and multiscale dynamics absent from available training data. We present SENDAI, a hierarchical **S**parse-measurement, **E**fficie**N**t **D**ata **A**ss**I**milation Framework that reconstructs full spatial states from hyper-sparse sensor observations by combining simulation-derived priors with learned discrepancy corrections. We demonstrate performance on satellite remote sensing, reconstructing MODIS-derived vegetation index fields across globally distributed sites. Using seasonal periods as a proxy for domain shift, the framework consistently outperforms established baselines that require substantially denser observations—SENDAI achieves substantial SSIM improvements over traditional baselines and recent high-frequency-based methods. These gains are particularly pronounced for landscapes with sharp boundaries and sub-seasonal dynamics; more importantly, the framework effectively preserves diagnostically relevant structures—such as field topologies, land cover discontinuities, and spatial gradients. The results highlight a lightweight and operationally viable framework for sparse-measurement reconstruction applicable to physically grounded inference, resource-limited deployment, and real-time monitoring and control.

## 1 INTRODUCTION

Reconstructing full spatiotemporal fields from sparse observations constitutes a fundamental challenge across Earth sciences, with applications spanning vegetation monitoring, hydrological modeling, and climate analysis (Weiss et al., 2020; Mohanty et al., 2017; Adrian et al., 2025; Jiang et al., 2025). Satellite remote sensing platforms such as MODIS provide unprecedented global coverage, yet cloud contamination, sensor gaps, and transmission constraints frequently yield incomplete spatial fields that compromise downstream analyses (Shen et al., 2015; Zhang et al., 2018).

Contemporary deep learning approaches have demonstrated impressive reconstruction capabilities but typically demand GPU clusters, massively labeled training datasets, and substantial computational resources that preclude deployment in operational or resource-constrained settings (He et al., 2016; Morel et al., 2025; Meraner et al., 2020; Cresson, 2018; Zhang et al., 2018). Recent works in physics-informed AI (Kutz et al., 2025; Karniadakis et al., 2021; Raissi et al., 2019; Fan et al., 2025; Liu et al., 2024) and neural operators (Lu et al., 2021; Li et al., 2020; Roy et al., 2025)

---

[*]Equal contribution

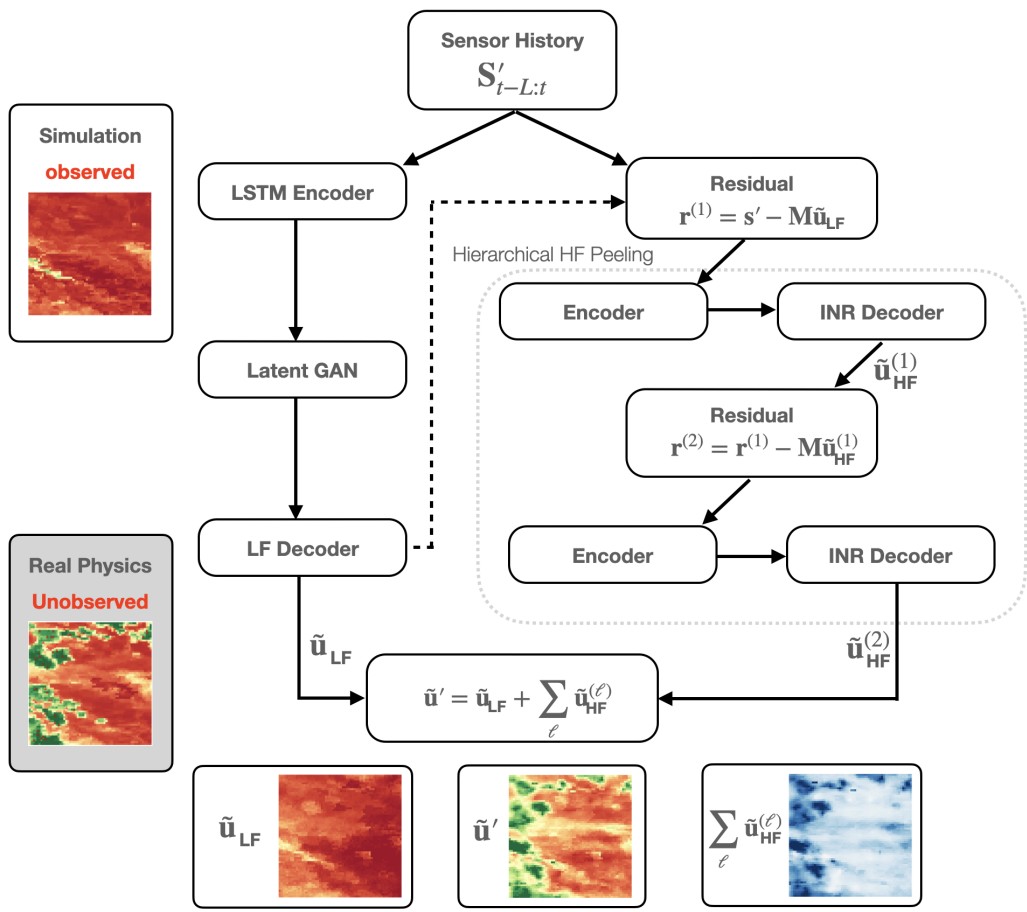

Figure 1: SENDAI architecture. The LF pathway learns dominant dynamics from simulation and aligns to ground truth via a latent GAN. The HF pathway employs sequential peeling layers to extract spectrally-distinct corrections.

have shown how physical structure can be embedded into surrogates for improved generalization. However, many practical reconstruction problems are not naturally posed as a well-specified PDE with reliable priors, motivating methods that remain effective under weak or unknown governing structure. Moreover, the assumption of stationarity is frequently violated in Earth observation contexts where phenological shifts, seasonal transitions, and land cover dynamics introduce substantial domain changes (Zeng et al., 2024; Truong et al., 2021; Cheng et al., 2025). Existing approaches often learn latent representations that blend multiple contributing effects, yielding entangled corrections with limited physical interpretability (Dylewsky et al., 2019; Wang et al., 2022; Chen et al., 2022).

In this work, we present SENDAI (**S**parse-measurement, **E**fficie**N**t **D**ata **A**ss**I**milation), a hierarchical data assimilation framework that reconstructs full spatial states from severely sparse sensor observations by combining simulation-derived priors with learned discrepancy corrections. The architecture decomposes reconstruction into two complementary pathways: (i) a *low-frequency pathway* that leverages Takens' embedding theorem (Takens, 2006) through shallow recurrent decoder networks (SHRED) (Williams et al., 2024) to capture dominant spatiotemporal dynamics, with latent-space adversarial alignment bridging distribution shifts; and (ii) a *high-frequency pathway* employing sequential frequency peeling with coordinate-based implicit neural representations (INRs) (Sitzmann et al., 2020; Tancik et al., 2020) to resolve fine-scale structure, sharp boundaries, and localized corrections.

The principal innovations of this work address three critical gaps in existing machine learning methods for spatiotemporal reconstruction:

1. *Extreme sparsity reconstruction.* The framework achieves effective full-state reconstruction from only 64 sensors covering 1.56% of the spatial domain—substantially below the density thresholds required by conventional methods.

2. *Computational efficiency for operational deployment.* The lightweight architecture enables training and inference on standard hardware within minutes, making it suitable for resource-constrained operational settings where extensive computational infrastructure is unavailable (Erichson et al., 2020).

3. *Hierarchical frequency peeling for heterogeneous fields.* We introduce a novel sequential peeling strategy that decomposes high-frequency corrections into interpretable layers with explicit spectral constraints and frequency exclusion mechanisms. Combined with coordinate-based INR decoders, this approach produces spatially coherent reconstructions that preserve topological structure.

While demonstrated here on vegetation index, the framework could be generalized beyond Earth observation to broader remote sensing tasks with learnable structure and sparse measurement on heterogeneous spatiotemporal fields. Code is available at: `https://github.com/xswzaqnjimko/SENDAI_framework`.

## 2 PRELIMINARIES

We demonstrate SENDAI on satellite remote sensing, reconstructing MODIS-derived NDVI fields across four globally distributed sites spanning Mediterranean, arid, and subtropical climates, using seasonal periods as proxies for the sim2real domain shift.

### 2.1 REMOTE SENSING DATA AND VEGETATION INDEX

We utilize imagery from the Moderate Resolution Imaging Spectroradiometer (MODIS) aboard NASA's Terra and Aqua satellites (Justice et al., 2002). Our primary state variable is the Normalized Difference Vegetation Index (NDVI), computed as:

$$\text{NDVI} = \frac{\rho_{\text{NIR}} - \rho_{\text{Red}}}{\rho_{\text{NIR}} + \rho_{\text{Red}}} \tag{1}$$

where $\rho_{\text{NIR}}$ and $\rho_{\text{Red}}$ denote surface reflectance in the near-infrared and red spectral bands, respectively (Tucker, 1979; Huete et al., 2002). Specifications of MODIS are in Appendix A.

### 2.2 DATA COLLECTION AND EXPERIMENTAL SETUP

**Data Acquisition and Processing.** All MODIS imagery was acquired through Google Earth Engine (Gorelick et al., 2017). For each study site, we define a 15 km × 15 km region resampled to a standardized 64 × 64 pixel grid. Complete specifications are provided in Appendix B.

**Study Sites.** We evaluate SENDAI across four globally distributed study sites: Central Valley (California, USA), Guadalquivir Valley (Spain), Imperial Valley (California, USA), and Tarim Basin (China). This geographic diversity demonstrates generalizability across heterogeneous landscapes. Table 2 summarizes site characteristics; details are provided in Appendix C.

**Seasonal Split Strategy and Sensor Configuration.** We treat observations from one seasonal period as "simulation" training data and observations from a different season as the "ground truth" to be reconstructed. Specifications are provided in Appendix D.

### 2.3 BASELINE METHODS

We compare against three established baselines widely used in remote sensing: ***SG+IDW*** (Savitzky-Golay filtering with Inverse Distance Weighting), ***HANTS+IDW*** (harmonic models with IDW), and ***Kriging*** (Gaussian Process regression with RBF kernel). We also evaluate ***MMGN*** (Luo et al., 2024), a recent implicit neural representation approach. All baselines operate with 64 sensors. Implementation details are in Appendix E.

## 3 SENDAI ARCHITECTURE

We develop a hierarchical data assimilation architecture that reconstructs full spatial fields from sparse sensor observations, building upon DA-SHRED and Cheap2Rich (Bao & Kutz, 2025a; Bao et al., 2026) while introducing: (i) a *sequential frequency peeling* strategy for interpretable decomposition; and (ii) a *coordinate-based INR* for spatially coherent reconstructions. The full field of the ground truth system is never observed during training. Figure 1 illustrates the complete pipeline.

### 3.1 PROBLEM FORMULATION

Let $\mathbf{u}_k = \mathbf{u}(\mathbf{x}, t_k) \in \mathbb{R}^n$ denote the full state at time $t_k$, where $n = H \times W$ is the spatial dimension. Sparse sensor measurements are given by $\mathbf{s}_k = \mathbf{M}\mathbf{u}_k \in \mathbb{R}^p$ with $p \ll n$, where $\mathbf{M} \in \mathbb{R}^{p \times n}$ is the sampling operator. Following the multiscale decomposition (Bao et al., 2026; Ilersich & Nair, 2025), we write:

$$\tilde{\mathbf{u}}'(t) = \tilde{\mathbf{u}}_{\text{LF}}(t) + \tilde{\mathbf{u}}_{\text{HF}}(t), \tag{2}$$

where $\tilde{\mathbf{u}}_{\text{LF}}$ captures dominant dynamics learned from simulation; $\tilde{\mathbf{u}}_{\text{HF}}$ represents fine-scale corrections.

### 3.2 LOW-FREQUENCY (LF) PATHWAY

The low-frequency pathway follows DA-SHRED (Bao & Kutz, 2025a), consisting of a temporal encoder trained on simulation data and a latent-space alignment mechanism.

**Temporal Encoder.** Given sensor time-history $\mathbf{S}'_{t-L:t} \in \mathbb{R}^{L \times p}$ with $L$ temporal lags, the encoder maps this sequence to a latent representation via a multi-layer LSTM:

$$\mathbf{z}_{\text{LF}}(t) = \mathcal{E}_{\text{LF}}(\mathbf{S}'_{t-L:t}; \theta_{\text{enc}}) = \text{LayerNorm}\left(\mathbf{h}_L^{(K)}\right), \tag{3}$$

where $\mathbf{h}_L^{(K)} \in \mathbb{R}^{d_z}$ is the final hidden state.

**Latent-Space GAN Alignment.** To address distribution mismatch, we employ a residual generator $\mathcal{G}$ that aligns latent distributions adversarially:

$$\tilde{\mathbf{z}}_{\text{LF}}(t) = \mathbf{z}_{\text{LF}}(t) + \gamma \cdot \mathcal{G}(\mathbf{z}_{\text{LF}}(t); \theta_{\mathcal{G}}), \tag{4}$$

where $\gamma$ is a learnable scale parameter.

**LF Decoder with Spectral Constraint.** The aligned latent code is decoded and low-pass filtered:

$$\tilde{\mathbf{u}}_{\text{LF}}(t) = \mathcal{P}_{k_c}\left(\mathcal{D}_{\text{LF}}(\tilde{\mathbf{z}}_{\text{LF}}(t); \theta_{\text{dec}})\right) \in \mathbb{R}^n, \tag{5}$$

where $\mathcal{P}_{k_c}$ retains only Fourier modes with wavenumber $k \leq k_c$.

### 3.3 HIERARCHICAL HIGH-FREQUENCY PEELING (HFP)

We introduce a *hierarchical peeling* structure that sequentially extracts frequency components layer by layer.

**Sequential Residual Computation.** Let $\mathbf{u}^{(0)} = \tilde{\mathbf{u}}_{\text{LF}}$ denote the low-frequency base reconstruction. For $\ell = 1, \ldots, N_{\text{peel}}$ hierarchical layers:

$$\mathbf{r}^{(\ell)}(t) = \mathbf{s}'(t) - \mathbf{M}\mathbf{u}^{(\ell-1)}(t), \tag{6}$$

$$\tilde{\mathbf{u}}_{\text{HF}}^{(\ell)}(t) = \mathcal{H}^{(\ell)}(\mathbf{r}^{(\ell)}(t); \theta_{\text{HF}}^{(\ell)}), \tag{7}$$

$$\mathbf{u}^{(\ell)}(t) = \mathbf{u}^{(\ell-1)}(t) + \tilde{\mathbf{u}}_{\text{HF}}^{(\ell)}(t). \tag{8}$$

**Frequency-Guided Sparsity with Exclusion.** Each peeling layer is trained with a bandlimited sparsity regularizer that *excludes* frequencies discovered by previous layers:

$$\mathcal{R}_{\text{sparse}}^{(\ell)} = \underbrace{\frac{\|\hat{\mathbf{u}}^{(\ell)}\|_{1,\mathcal{B}}}{\|\hat{\mathbf{u}}^{(\ell)}\|_{2,\mathcal{B}} + \epsilon}}_{\text{in-band L1/L2}} + \beta_1 \mathcal{P}_{\bar{\mathcal{B}}}^{(\ell)} + \beta_2 \mathcal{P}_{\mathcal{E}}^{(\ell)}, \tag{9}$$

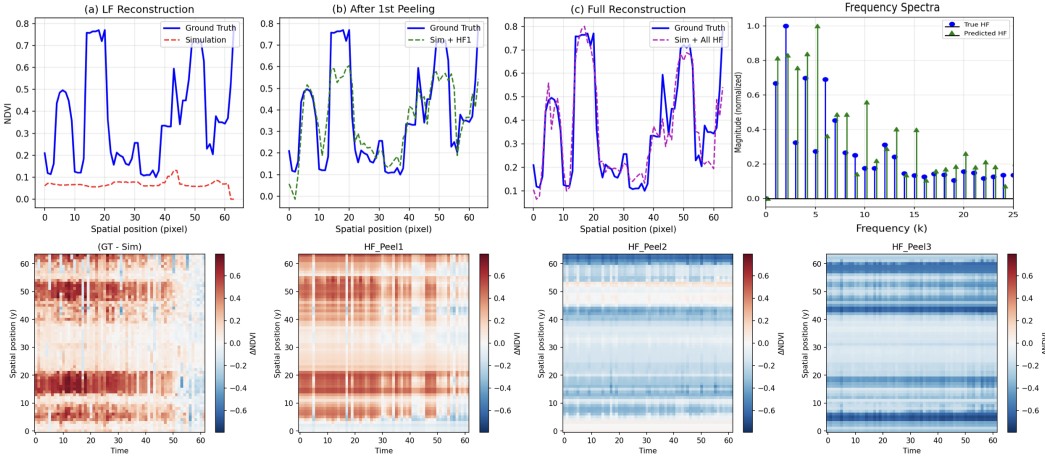

Figure 2: Hierarchical frequency peeling on a 1D NDVI slice from the Tarim Basin site. Top row presents single time-point reconstruction; bottom row presents spatiotemporal HF decomposition.

where $\mathcal{B}$ is the target frequency band, $\bar{\mathcal{B}}$ its complement, and $\mathcal{E}^{(\ell)}$ is the union of exclusion regions around frequencies discovered by previous layers.

**Adaptive Top-$k_\ell$ Mode Selection.** For each peeling layer, we employ adaptive selection via spectral analysis of the input residual. Details are provided in Appendix F.4.

### 3.4 COORDINATE-BASED IMPLICIT NEURAL REPRESENTATION

Direct MLP-based mapping often produces "dotted" artifacts at sensor locations. We replace the direct MLP decoder with a *coordinate-based INR* (Sitzmann et al., 2020; Tancik et al., 2020) that learns a continuous function over space.

**Architecture.** Each HF peeling layer $\mathcal{H}^{(\ell)}$ consists of: (i) a *Sensor Residual Encoder* mapping the $p$-dimensional residual to a latent code $\mathbf{z}_{\text{HF}}^{(\ell)} \in \mathbb{R}^{d_{\text{HF}}}$; and (ii) a *Coordinate-Based Decoder* computing for each spatial coordinate $(x, y)$:

$$u_{\text{HF}}^{(\ell)}(x, y) = \gamma^{(\ell)} \cdot \mathcal{D}_{\text{INR}}^{(\ell)}\big([\text{PE}(x, y); \mathbf{z}_{\text{HF}}^{(\ell)}]\big), \tag{10}$$

where $\text{PE}(\cdot)$ is a Fourier positional encoding. Details on spatial smoothness regularization and the full HF layer loss are in Appendix F.5.

### 3.5 TRAINING PIPELINE

Training proceeds in three stages: ***Stage 1*** trains the base SHRED model on simulation data; ***Stage 2*** trains latent alignment via adversarial learning; ***Stage 3*** performs hierarchical HF peeling with adaptive $k_\ell$ selection. Full details are in Appendix F.

## 4 RESULTS

We evaluate SENDAI across four geographically diverse study sites. All experiments employ 64 sensors ($\sim$1.5% of pixels). We adopt SSIM as our primary metric, as it captures structural preservation that RMSE alone cannot assess.

### 4.1 SYNTHETIC ANALYSIS

We validate the HFP methodology on a 1D slice extracted from real MODIS NDVI data where simulation and ground truth exhibit clear spectral discrepancy. Additional synthetic validation on a traveling wave system is provided in Appendix G.

**NDVI 1D Validation.** To validate hierarchical peeling on remote sensing data with inherent noise and non-stationary dynamics, we extract a 1D slice from the Tarim Basin site (Figure 7). Despite the HF

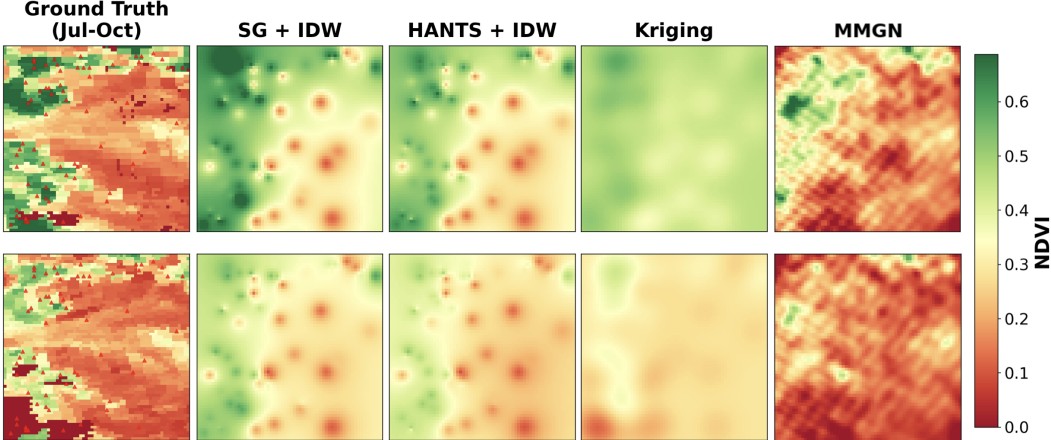

Figure 3: Baseline reconstruction comparison for the Tarim Basin site. Red markers indicate sensor locations.

residual exhibiting widespread energy, hierarchical peeling successfully captures the majority of the modes (Figure 2) while producing interpretable components corresponding to distinct spatiotemporal scales. The first layer ($HF_1$) captures coherent spatio-temporal variability consistent with climate-driven phenological offsets, while subsequent layers isolate time-invariant spatial gradients suggestive of persistent hydrological and edaphic controls. Success on data without clean sinusoidal patterns demonstrates robustness to realistic data imperfections.

**Advantages of Hierarchical Peeling.** The experiments reveal several key advantages: (i) robustness to non-ideal signals; (ii) frequency exclusion preventing mode leakage; (iii) stable training with sequential convergence; and (iv) interpretability with nested decompositions mapping onto scale-dependent control mechanisms.

## 4.2 BASELINE PERFORMANCE AND SENDAI RECONSTRUCTION

We establish baseline performance and evaluate both SENDAI variants: (i) SENDAI Jr. for sites with primarily low-frequency discrepancies (Central Valley, Guadalquivir Valley), and (ii) full SENDAI with hierarchical HFP for sites requiring high-frequency correction (Imperial Valley, Tarim Basin).

Figure 3 illustrates baseline reconstruction quality on the Tarim Basin site, which presents challenging heterogeneous spatial structure. A critical observation emerges: *baseline methods fundamentally fail to preserve topological structure despite achieving moderate RMSE values*. IDW-based methods exhibit "bullseye" artifacts; Kriging produces over-smoothed reconstructions destroying sharp boundaries. MMGN exhibits comparable limitations under our extreme sparsity regime.

Table 1 summarizes performance across all four sites. For SENDAI Jr. sites, the framework achieves SSIM improvements of +120% (Central Valley) and +98% (Guadalquivir Valley) over the best baselines, demonstrating successful preservation of field boundaries and vegetation gradients. Figure 4b presents reconstruction for Central Valley, showing preservation of essential topological structure.

For sites requiring high-frequency correction, the full SENDAI hierarchical architecture achieves SSIM of $0.4668 \pm 0.0390$ (Imperial Valley) and $0.4777 \pm 0.0205$ (Tarim Basin), representing SSIM gains of 15.5% and 36.3% respectively, over SENDAI Jr. The Tarim Basin site exhibits the most dramatic improvement, where sharp mountain-basin boundaries require high-frequency corrections that smooth decoders cannot capture.

Figure 5 presents the full hierarchical reconstruction for the Tarim Basin site. While all baseline methods fail to resolve sharp boundaries, SENDAI successfully reconstructs these landscape features. The learned HF component exhibits coherent spatial structure aligned with boundaries, confirming that peeling layers discover physically meaningful corrections. Sensor sensitivity analysis is provided in Figure 4a. Additional ablation studies examining sensitivity to temporal lag, maximum target

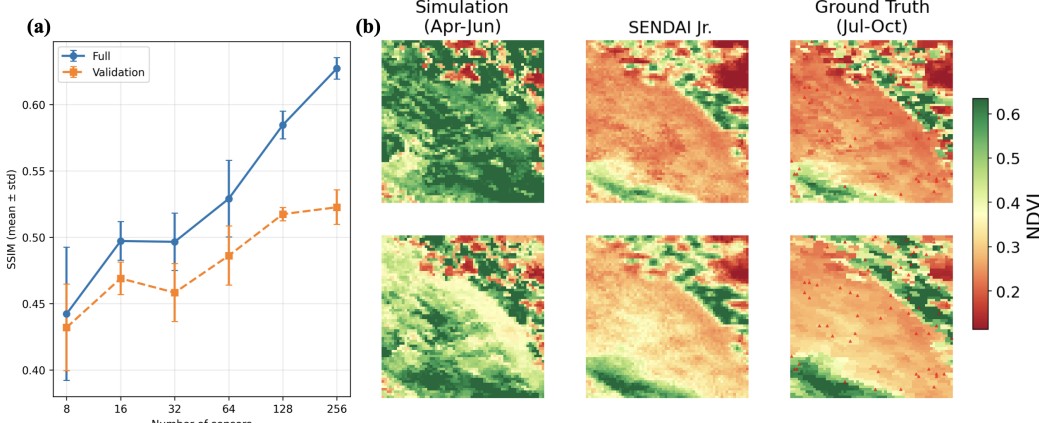

Figure 4: (a) Sensor sensitivity results on the Tarim Basin site for the full SENDAI framework. (b) SENDAI Jr. reconstruction for the Central Valley site.

Table 1: SENDAI reconstruction performance compared with baselines. Left: SENDAI Jr. sites. Right: Full SENDAI sites. Best results per site are **bolded**. All results are averaged over five independent runs.

| | | SENDAI Jr. Central Valley | Guadalquivir | | | SENDAI Imperial Valley | Tarim Basin |
|---|---|---|---|---|---|---|---|
| Sim. vs. GT | RMSE | 0.1965 | 0.2387 | Sim. vs. GT | RMSE | 0.2157 | 0.2077 |
| | SSIM | 0.4751 | 0.2464 | | SSIM | 0.3488 | 0.3448 |
| SG + IDW | RMSE | $0.1447 \pm 0.0029$ | $\mathbf{0.1444} \pm 0.0023$ | SG + IDW | RMSE | $0.1599 \pm 0.0054$ | $0.1794 \pm 0.0011$ |
| | SSIM | $0.2612 \pm 0.0098$ | $0.1849 \pm 0.0084$ | | SSIM | $0.1123 \pm 0.0146$ | $0.1308 \pm 0.0320$ |
| HANTS + IDW | RMSE | $0.1451 \pm 0.0029$ | $0.1496 \pm 0.0022$ | HANTS + IDW | RMSE | $0.1603 \pm 0.0054$ | $0.1806 \pm 0.0010$ |
| | SSIM | $0.2504 \pm 0.0096$ | $0.1755 \pm 0.0083$ | | SSIM | $0.1049 \pm 0.0145$ | $0.1214 \pm 0.0315$ |
| Kriging | RMSE | $0.1634 \pm 0.0027$ | $0.1481 \pm 0.0008$ | Kriging | RMSE | $0.1591 \pm 0.0024$ | $0.2163 \pm 0.0053$ |
| | SSIM | $0.0922 \pm 0.0132$ | $0.0878 \pm 0.0028$ | | SSIM | $0.0916 \pm 0.0190$ | $0.0449 \pm 0.0174$ |
| | | | | MMGN | RMSE | $0.1786 \pm 0.0076$ | $0.1783 \pm 0.0108$ |
| | | | | | SSIM | $0.0778 \pm 0.0270$ | $0.1226 \pm 0.0448$ |
| | | | | SENDAI Jr. | RMSE | $0.1708 \pm 0.0029$ | $0.1827 \pm 0.0046$ |
| | | | | | SSIM | $0.4041 \pm 0.0248$ | $0.3505 \pm 0.0269$ |
| **SENDAI Jr.** | RMSE | $\mathbf{0.1068} \pm 0.0015$ | $0.1474 \pm 0.0014$ | **SENDAI** | RMSE | $\mathbf{0.1486} \pm 0.0063$ | $\mathbf{0.1208} \pm 0.0109$ |
| | SSIM | $\mathbf{0.5747} \pm 0.0478$ | $\mathbf{0.3655} \pm 0.0070$ | | SSIM | $\mathbf{0.4668} \pm 0.0390$ | $\mathbf{0.4777} \pm 0.0205$ |
| SSIM Improvement | | +120.0% | +97.7% | SSIM Improvement | | +15.5% | +36.3% |

frequency, and sensor noise robustness are provided in Appendix J. These experiments confirm that SENDAI's performance is robust across a range of hyperparameter settings.

## 4.3 COMPUTATIONAL EFFICIENCY

SENDAI is remarkably efficient computationally. Unlike contemporary deep learning approaches requiring GPU clusters (Meraner et al., 2020; Cresson, 2018), SENDAI operates on standard CPU hardware with training times measured in minutes per site. This efficiency derives from the shallow architecture and exploitation of Takens' embedding theorem.

## 5 BROADER IMPLICATIONS FOR EARTH OBSERVATION AND REMOTE SENSING

While the preceding sections demonstrated SENDAI's efficacy for NDVI reconstruction, the methodological contributions extend substantially beyond this specific application. This section examines the broader implications of our framework for Earth observation science, operational remote sensing systems, and emerging challenges in spatiotemporal field reconstruction.

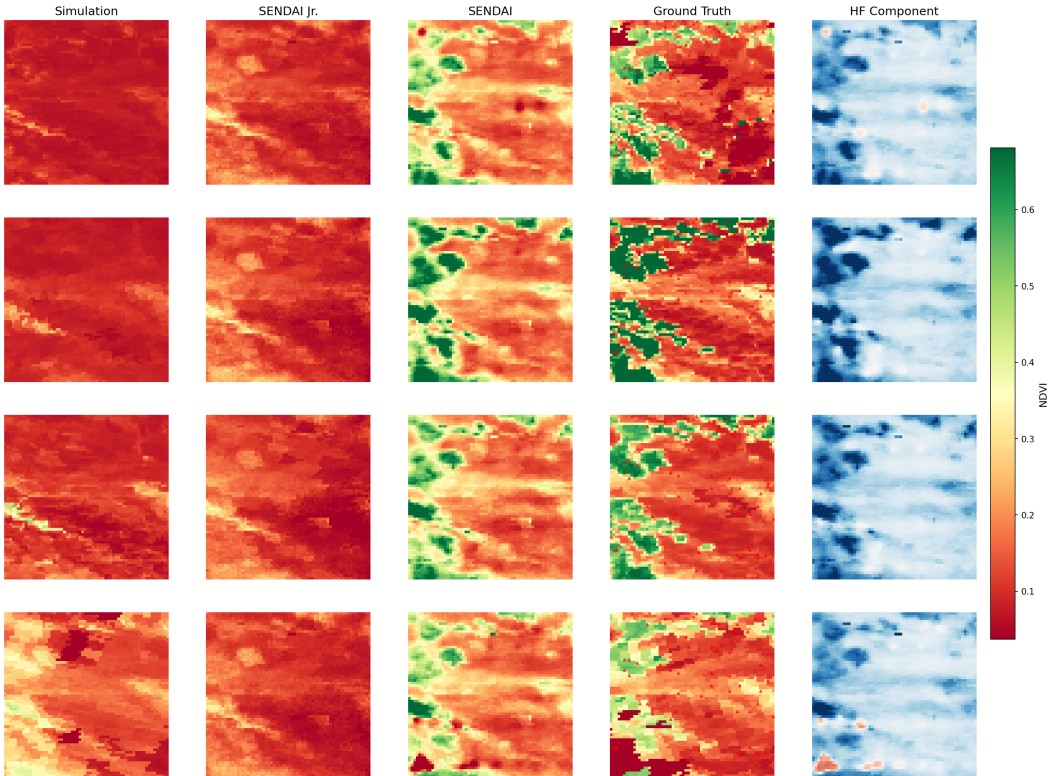

Figure 5: Full SENDAI hierarchical multiscale DA-SHRED reconstruction for the Tarim Basin site.

## 5.1 DOMAIN ADAPTATION AND MULTI-DOMAIN TRANSFERABILITY

A fundamental contribution of this work lies in demonstrating effective *domain adaptation* for spatiotemporal field reconstruction—the capacity to transfer learned representations from one observational regime to another with minimal target-domain supervision. Our seasonal split strategy constitutes a form of simulation-to-reality (sim2real) transfer (Truong et al., 2021; Josifovski et al., 2025) applicable across Earth observation variables where distributional shifts arise from sensor degradation, cross-sensor harmonization, or temporal non-stationarity (Zeng et al., 2024). SENDAI's adversarial latent-space alignment offers an alternative to traditional supervised approaches that require prohibitive amounts of dense labeled data: learn the *structure* of plausible spatial configurations from data-rich periods, then adapt to data-sparse regimes using only sensor-location observations.

While this study focused on NDVI reconstruction, the framework's architecture is agnostic to the specific variable of interest. The essential requirements are: (i) a spatiotemporally coherent field exhibiting learnable structure, (ii) availability of a reference dataset (simulation, historical observations, or cross-sensor data) from which to learn spatial priors, and (iii) sparse observations from the target domain for adaptation. These requirements are satisfied across numerous Earth observation applications: soil moisture exhibits strong spatial structure governed by topography, soil properties, and precipitation patterns (Mohanty et al., 2017); thermal imagery for land surface temperature suffers from cloud contamination analogous to optical data (Li et al., 2013); seasonal snow dynamics exhibit strong spatial structure with microwave observations providing all-weather sensing (Lievens et al., 2019); and flood inundation extent changes rapidly with SAR observations penetrating clouds (Schumann & Moller, 2015). For each application, the SENDAI framework offers a consistent methodological template: establish spatial priors from data-rich sources, adapt to observation-sparse target conditions via latent-space alignment, and employ hierarchical frequency decomposition when fine-scale corrections are required.

## 5.2 Hardware Efficiency: Storage, Bandwidth, and Latency

The demonstrated capacity to reconstruct full spatiotemporal fields from sparse observations (64 sensors representing 1.56% of pixels) suggests novel paradigms for satellite data systems addressing storage, bandwidth, and latency constraints. Rather than transmitting complete imagery with conventional compression (Xie et al., 2021; Faria et al., 2012), an alternative paradigm involves transmitting only sparse sensor measurements alongside periodically updated model weights, achieving substantial data reduction while preserving spatial structure essential for downstream analysis.

This paradigm addresses three interconnected challenges. First, **bandwidth reduction**: for resource-constrained regions or deep-space exploration missions where communication bandwidth severely constrains scientific return (De Cola et al., 2011). Second, **storage efficiency**: reducing onboard storage requirements by maintaining only sparse measurements and model parameters. Third, **low-latency decision support**: generating provisional reconstructions from partial observations before complete imagery becomes available, critical for disaster response and precision agriculture timing decisions (Denby & Lucia, 2020; Giuffrida et al., 2020). The implications for operational monitoring systems are substantial—near-real-time agricultural monitoring requires rapid processing of incoming satellite data streams, and SENDAI's short inference times per scene suggest a viable path toward high-quality reconstruction within operational latency constraints.

## 6 Conclusions and Future Directions

The experimental results show that SENDAI robustly reconstructs heterogeneous spatiotemporal NDVI fields across diverse geographic and climatic settings, adapting from simulation to ground-truth observations across seasonal boundaries while recovering mesoscale patterns and fine-scale structure via hierarchical peeling. Performance depends on landscape complexity: SENDAI Jr. suffices for smoother fields with persistent spatial structure, whereas sites with sharp boundaries and sub-seasonal dynamics benefit from the full SENDAI framework with hierarchical structure.

Several limitations and future directions remain. First, SENDAI assumes stationary spatial structure between simulation and ground truth periods, which may break under land cover change, management shifts, or disturbances; extending the framework to non-stationary settings is an important next step. Second, our current random sensor placement could be improved through information-theoretic design (Bao & Kutz, 2025b; Santos et al., 2023), potentially reducing sensor requirements. Third, although we observed transferability across regions, rigorous cross-continent generalization tests are still needed to distinguish universal from region-specific representations. Finally, extending to multivariate reconstruction (e.g., NDVI, land surface temperature, soil moisture) could leverage cross-variable correlations via the shared latent space.

## Impact Statement

In this paper we investigate the applications of Machine Learning to the frontier of remote sensing. The SENDAI framework enables accurate heterogeneous spatiotemporal reconstruction and domain adaptation from severely sparse sensor observations, with potential benefits for environmental monitoring, space exploration, disaster response, and climate analysis, particularly in resource-constrained areas where dense observations are expensive or unavailable.

## Acknowledgments

This work was supported in part by the US National Science Foundation (NSF) AI Institute for Dynamical Systems (dynamicsai.org), grant 2112085. JNK further acknowledges support from the Air Force Office of Scientific Research (FA9550-24-1-0141).

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

## A   MODIS PLATFORM DETAILS

The Moderate Resolution Imaging Spectroradiometer (MODIS) is a key instrument aboard NASA's Terra and Aqua satellites, launched in 1999 and 2002 respectively, providing continuous global observations for over two decades (Justice et al., 2002). MODIS acquires data in 36 spectral bands ranging from 0.405 to 14.385 $\mu$m, with spatial resolutions of 250 m, 500 m, and 1000 m depending on the band. The instrument's 2330 km viewing swath width enables complete global coverage every one to two days, making it uniquely suited for monitoring dynamic land surface processes at regional to continental scales.

The Terra satellite follows a descending sun-synchronous orbit with a 10:30 AM local equatorial crossing time, while Aqua follows an ascending orbit with a 1:30 PM crossing. This complementary configuration provides up to two observations per day for any given location, significantly increasing the probability of obtaining cloud-free imagery. In this study, we utilize the Collection 6.1 daily surface reflectance products (MOD09GA from Terra and MYD09GA from Aqua), which provide atmospherically corrected surface reflectance values processed using the Second Simulation of the Satellite Signal in the Solar Spectrum (6S) radiative transfer model.

The Normalized Difference Vegetation Index (NDVI) exploits the distinctive spectral signature of photosynthetically active vegetation, which strongly absorbs red light for photosynthesis while reflecting near-infrared (NIR) radiation due to leaf cellular structure (Tucker, 1979). NDVI values typically range from approximately $-0.1$ for water bodies and bare soil to $0.8$–$0.9$ for dense, healthy vegetation, responding sensitively to changes in chlorophyll content, leaf area, and vegetation fraction (Huete et al., 2002).

## B   DATA PROCESSING PIPELINE DETAILS

All MODIS imagery was acquired through Google Earth Engine (GEE) (Gorelick et al., 2017), a cloud-based platform providing direct access to petabyte-scale geospatial archives. For each study site, imagery from both Terra and Aqua sensors was merged to maximize temporal density, yielding up to two potential observations per day.

Cloud contamination is addressed using the `state_1km` quality assurance band, retaining only pixels flagged as "clear" (bit pattern 00 in bits 0–1). Images with less than 70% valid pixel coverage are excluded to ensure spatial coherence. For each study site, we define a 15 km $\times$ 15 km square region centered on representative coordinates, resampled to a standardized $64 \times 64$ pixel grid at fine resolution. This fixed grid dimension facilitates consistent architecture across study sites and enables direct comparison of model performance.

To ensure consistent temporal density while accounting for variable cloud cover, we employ an equally-spaced sampling strategy targeting 70–90 valid images per approximately 90-day observation period. In practice, persistent cloud cover at several sites limited acquisition to 40–50 valid images per period, testing the framework's robustness under reduced temporal sampling.

For each study site, the data generation pipeline produces simulation data (from one seasonal period) and ground truth data (from a different seasonal period). The simulation data serves to train the base SHRED architecture, learning a latent representation of the state space structure. The ground truth data provides sensor-only observations for the SENDAI, where the model learns to adapt its latent space to the distributional shift between seasons while reconstructing full-state fields from sparse measurements.

## C   STUDY SITE DESCRIPTIONS

### C.1   WESTERN UNITED STATES: CENTRAL VALLEY, CALIFORNIA

The Central Valley represents one of the world's most productive agricultural regions, characterized by Mediterranean climate (Köppen Csa) with hot, dry summers and mild, wet winters. The landscape is dominated by irrigated permanent crops (orchards, vineyards) and annual field crops. The simulation period (April–June) captures spring green-up and early crop development, while the ground truth period (July–October) spans peak summer productivity through early senescence. The

Table 2: Study site characteristics and observation periods. Sites are grouped by the architecture variant employed: SENDAI Jr. (simpler seasonal transitions) or SENDAI (complex phenological dynamics requiring high-frequency correction). All data are from 2023. Climate classifications follow the Köppen system: Csa = hot-summer Mediterranean; BWh = hot desert; BWk = cold desert.

| Region ID | Location Name | Center (Lon, Lat) | Climate | Land Cover | Sim. | GT | Model |
|---|---|---|---|---|---|---|---|
| *North America* | | | | | | | |
| western_us | Central Valley, CA | $(-120.5, 36.5)$ | Csa | Irrigated cropland | Apr–Jun | Jul–Oct | SENDAI Jr. |
| southwestern_us | Imperial Valley, CA | $(-115.5, 32.8)$ | BWh | Irrigated cropland | Apr–Jun | Jul–Oct | SENDAI |
| *Europe* | | | | | | | |
| western_spain | Guadalquivir Valley | $(-6.25, 37.25)$ | Csa | Mixed agriculture | Feb–Apr | Sep–Dec | SENDAI Jr. |
| *Asia* | | | | | | | |
| northwestern_china | Tarim Basin | $(83.5, 41.5)$ | BWk | Oasis agriculture | Apr–Jun | Jul–Oct | SENDAI |

strong phenological signal and relatively cloud-free conditions make this site suitable for the standard SENDAI Jr. architecture.

## C.2 Southwestern United States: Imperial Valley, California

Imperial Valley presents an extreme hot desert environment (Köppen BWh) where agriculture is entirely dependent on irrigation from the Colorado River. The sharp contrast between verdant irrigated fields and surrounding bare desert creates pronounced spatial heterogeneity. Multiple cropping cycles per year and variable irrigation schedules introduce high-frequency temporal dynamics requiring the full SENDAI architecture to capture sub-seasonal variability.

## C.3 Western Spain: Lower Guadalquivir Valley

The Guadalquivir Valley exhibits Mediterranean climate (Köppen Csa) with a distinctive reversed phenological calendar compared to northern hemisphere temperate regions—vegetation green-up occurs during mild, wet winters rather than spring. The February–April simulation period captures peak winter greenness and early spring drying, while September–December represents the onset of the growing season following summer drought. This phenological inversion provides a critical test of domain adaptation capability.

## C.4 Northwestern China: Tarim Basin

The Tarim Basin represents a hyper-arid cold desert environment (Köppen BWk) surrounded by the Tianshan and Kunlun mountain ranges. Agriculture is concentrated in narrow oasis strips fed by glacial meltwater, creating extreme spatial gradients between irrigated fields and surrounding desert. The April–June simulation period captures spring irrigation onset and crop establishment, while July–October encompasses peak productivity. The localized nature of vegetation and strong background contrasts require hierarchical frequency decomposition.

## D Seasonal Split Strategy and Sparse Sensor Configuration Details

Table 2 presents study site characteristics and observation periods, motivated by phenological domain shift. Together, they exhibit phenological coherence but heterogeneous spatiotemporal transitions in NDVI: vegetation dynamics vary not only across space but also exhibit distinct temporal autocorrelation structures between seasons (due to land cover heterogeneity, soil moisture gradients, and management practices, etc). For Mediterranean climates with reversed seasonality, the seasonal splits capture analogous phenological transitions adapted to local climate rhythms. The SENDAI framework must therefore learn to reconstruct fields that are simultaneously spatially heterogeneous and temporally non-stationary.

A central tenet of the SENDAI architecture is that full-state reconstruction can be achieved from sparse temporal observations at a limited number of spatial locations, leveraging Takens' embedding

theorem and learned decoder representations (Williams et al., 2024; Bao & Kutz, 2025a). We employ $p = 64$ randomly placed sensors, representing 1.56% of the full state space ($64 \times 64 = 4096$ pixels)—comparable to or lower than typical ground-based monitoring network densities. Sensor locations are randomly selected, excluding boundary pixels (2-pixel buffer). Following the SHRED formulation, sensor observations are organized into time-delay embeddings of length $L = 5$ lags, where at each time step $t$, the input to the encoder consists of the sensor history $\mathbf{S}_t = [\mathbf{s}_{t-L+1}, \ldots, \mathbf{s}_t]^\top \in \mathbb{R}^{L \times p}$, enabling the model to infer temporal derivatives and phenological trends critical for tracking dynamic vegetation changes.

## E  BASELINE IMPLEMENTATION DETAILS

### E.1  SG + IDW IMPLEMENTATION

Savitzky-Golay filtering is applied with window length 7 and polynomial order 2, providing local temporal smoothing while preserving phenological trends. Inverse Distance Weighting uses power parameter $p = 2$ (standard inverse-square weighting). This baseline represents the computational floor—no training required, sub-second inference.

### E.2  HANTS + IDW IMPLEMENTATION

HANTS fits a truncated Fourier series with 3 harmonic terms (annual, semi-annual, and quarterly cycles) at each sensor location. The fitting procedure uses iterative refinement to reject cloud-contaminated observations. Spatial interpolation follows the same IDW protocol as SG+IDW.

### E.3  KRIGING IMPLEMENTATION

Gaussian Process regression employs an RBF (squared exponential) kernel with automatic relevance determination. For Kriging-GT, lengthscale and variance hyperparameters are optimized via maximum likelihood on the ground truth sensor observations at each timestep. For Kriging-Sim, hyperparameters are pre-fitted on simulation full fields and held fixed during ground truth reconstruction, providing an intentionally favorable setting analogous to the use of simulation for encoder-decoder pretraining.

### E.4  MMGN IMPLEMENTATION

The Multiplicative and Modulated Gabor Network (MMGN) (Luo et al., 2024) represents a recent advance in implicit neural representations (INRs) for continuous field reconstruction from sparse observations. We implement MMGN following the original architecture and training protocol to provide a rigorous comparison with neural network-based approaches.

**Architecture.**  MMGN employs an auto-decoder architecture that learns a continuous spatial representation conditioned on per-timestep latent codes. The decoder consists of multiplicative layers combining Gabor filters with bilinear fusion of coordinates and latent information:

$$\mathbf{h}^{(0)} = g_0(\mathbf{x}) \odot \mathcal{F}_0(\mathbf{0}, \mathbf{z}), \quad \mathbf{h}^{(\ell+1)} = g_{\ell+1}(\mathbf{x}) \odot \mathcal{F}_{\ell+1}(\mathbf{h}^{(\ell)}, \mathbf{z}), \tag{11}$$

where $g_\ell(\mathbf{x})$ are Gabor filters applied to spatial coordinates, $\mathcal{F}_\ell$ are bilinear fusion layers, and $\mathbf{z} \in \mathbb{R}^{d_z}$ is a learnable latent code for each time instance. The Gabor filters take the form:

$$g_\ell(\mathbf{x}) = \sin(\mathbf{W}_g \mathbf{x} + \mathbf{b}_g) \odot \exp\left(-\frac{\gamma}{2}\|\mathbf{x} - \boldsymbol{\mu}\|^2\right), \tag{12}$$

with learnable centers $\boldsymbol{\mu}$ and bandwidth parameters $\gamma$ sampled from a Gamma distribution. This multiplicative structure enables the network to represent the output as a linear combination of Gabor basis functions, providing shift-invariance properties beneficial for spatial field reconstruction.

**Adaptation to Our Setting.**  The original MMGN was evaluated on climate simulation data (CESM2 global surface temperature, $192 \times 288$ resolution, 1024 timesteps) and satellite sea surface temperature (GHRSST, $901 \times 1001$ resolution, 360 timesteps) with sensor coverage ranging from 5% to 50%. Our MODIS NDVI reconstruction task presents four key differences: (i) substantially sparser

observations (1.56% vs. 5–50%), (ii) smaller spatial extent ($64 \times 64$ pixels), (iii) significantly shorter temporal sequences ($\sim$70 timesteps vs. 360–1024), and (iv) heterogeneous agricultural landscapes with sharp boundaries rather than smoothly varying oceanographic or atmospheric fields.

We apply MMGN with identical sensor configurations (64 randomly placed sensors) and temporal splits as all other methods. The model learns separate latent codes for each timestep, with training performed on sensor observations only and evaluation on full-field reconstruction.

# F SENDAI ARCHITECTURE DETAILS

This appendix provides complete architectural specifications for the SENDAI components discussed in Section 3.

## F.1 BASE SHRED MODEL

The base SHRED architecture consists of an LSTM temporal encoder and an MLP spatial decoder.

**LSTM Encoder.** The encoder processes sensor time-histories $\mathbf{S}_{t-L:t} \in \mathbb{R}^{L \times p}$:

$$\mathbf{h}_\tau, \mathbf{c}_\tau = \mathrm{LSTM}(\mathbf{s}_\tau, \mathbf{h}_{\tau-1}, \mathbf{c}_{\tau-1}), \quad \tau = t - L + 1, \ldots, t, \tag{13}$$

with $K$ stacked LSTM layers (typically $K = 2$), hidden dimension $d_z$, and dropout applied between layers during training. The latent representation is:

$$\mathbf{z} = \mathrm{LayerNorm}(\mathbf{h}_t^{(K)}), \tag{14}$$

where $\mathbf{h}_t^{(K)}$ is the final hidden state of the top layer.

**MLP Decoder.** The decoder maps latent codes to full spatial states:

$$\mathcal{D}_{\mathrm{LF}}(\mathbf{z}) = \mathbf{W}_D \cdot \mathrm{ReLU}(\mathrm{LN}(\mathbf{W}_{D-1} \cdots \mathrm{ReLU}(\mathrm{LN}(\mathbf{W}_1 \mathbf{z} + \mathbf{b}_1)) \cdots)), \tag{15}$$

where LN denotes layer normalization and the hidden layer dimensions are specified in Table 3.

## F.2 DA-SHRED LATENT TRANSFORM

The latent transformation module adapts simulation-trained representations to ground truth sensor data:

$$\mathcal{T}(\mathbf{z}) = \tanh\left(\mathbf{W}_2 \cdot \mathrm{ReLU}(\mathbf{W}_1 \mathbf{z} + \mathbf{b}_1) + \mathbf{b}_2\right), \tag{16}$$

with the transformed latent given by Eq. equation 4. The $\tanh$ nonlinearity bounds the correction magnitude, and the learnable scale $\gamma$ is initialized to 0.1.

**GAN Discriminator.** The discriminator $\mathcal{D} : \mathbb{R}^{d_z} \to [0, 1]$ is a 3-layer MLP with LeakyReLU activations (slope 0.2):

$$\mathcal{D}(\mathbf{z}) = \sigma\left(\mathbf{W}_3 \cdot \mathrm{LReLU}(\mathbf{W}_2 \cdot \mathrm{LReLU}(\mathbf{W}_1 \mathbf{z}))\right), \tag{17}$$

where $\sigma$ is the sigmoid function. Training uses the binary cross-entropy objectives:

$$\mathcal{L}_D = -\mathbb{E}_{\mathbf{z} \sim p_{\mathrm{gt}}}[\log \mathcal{D}(\mathbf{z})] - \mathbb{E}_{\mathbf{z} \sim p_{\mathrm{sim}}}[\log(1 - \mathcal{D}(\mathcal{G}(\mathbf{z})))], \tag{18}$$

$$\mathcal{L}_G = -\mathbb{E}_{\mathbf{z} \sim p_{\mathrm{sim}}}[\log \mathcal{D}(\mathcal{G}(\mathbf{z}))]. \tag{19}$$

**SENDAI Jr. Pipeline.** For datasets where the sim2real gap is primarily low-frequency, a simplified two-stage pipeline suffices. Stage 1 trains the LSTM encoder and MLP decoder on simulation data with either sparse sensor histories or full-state supervision until convergence. Stage 2 then encodes both simulation and ground truth sensor data to obtain latent distributions and trains the GAN components (generator $\mathcal{G}$ and discriminator $\mathcal{D}$) on these latent codes to align the distributions. This variant relies solely on latent-space alignment to bridge the sim2real gap and is recommended as a baseline before deploying the full hierarchical architecture. The full pipeline should be used when spectral analysis of post-alignment residuals reveals significant high-frequency structure, evidenced by distinct peaks in the FFT magnitude spectrum of $\mathbf{s}'(t) - \mathbf{M}\tilde{\mathbf{u}}_{\mathrm{LF}}(t)$.

### F.3 COORDINATE-BASED INR FOR HF CORRECTION

The implicit neural representation for high-frequency correction consists of an encoder and coordinate-based decoder.

**Sensor Residual Encoder.** Maps $p$-dimensional residuals to latent codes:

$$\mathcal{E}_{\text{HF}}(\mathbf{r}) = \mathbf{W}_E^{(2)} \cdot \text{ReLU}(\text{LN}(\mathbf{W}_E^{(1)}\mathbf{r} + \mathbf{b}_E^{(1)})) + \mathbf{b}_E^{(2)}, \tag{20}$$

with output dimension $d_{\text{HF}}$ (typically 64).

**Fourier Positional Encoding.** For coordinates $(x, y) \in [0, 1]^2$ and $L$ frequency bands with log-spaced frequencies $\sigma_j = 2^{(j-1)\log_2 \sigma_{\max}/(L-1)}$:

$$\text{PE}(x, y) = [x, y, \sin(2\pi\sigma_1 x), \cos(2\pi\sigma_1 x), \sin(2\pi\sigma_1 y), \cos(2\pi\sigma_1 y), \ldots] \in \mathbb{R}^{2+4L}. \tag{21}$$

Typical values are $L = 16$ frequency bands with $\sigma_{\max} = 8.0$.

**Coordinate Decoder.** The decoder MLP takes the concatenation $[\text{PE}(x, y); \mathbf{z}_{\text{HF}}] \in \mathbb{R}^{2+4L+d_{\text{HF}}}$:

$$\mathcal{D}_{\text{INR}}([\text{PE}; \mathbf{z}]) = \mathbf{W}_3 \cdot \text{ReLU}(\text{LN}(\mathbf{W}_2 \cdot \text{ReLU}(\text{LN}(\mathbf{W}_1[\text{PE}; \mathbf{z}])))). \tag{22}$$

The final layer produces a scalar output for each queried coordinate.

**Batched Coordinate Queries.** At inference, all $n = H \times W$ grid coordinates are queried simultaneously. For memory efficiency with large grids, coordinates are processed in chunks:

$$u_{\text{HF}}(i, j) = \gamma \cdot \mathcal{D}_{\text{INR}}([\text{PE}(i/H, j/W); \mathbf{z}_{\text{HF}}]), \quad \forall (i, j) \in \{0, \ldots, H-1\} \times \{0, \ldots, W-1\}. \tag{23}$$

### F.4 2D FREQUENCY SPARSITY REGULARIZATION

For 2D spatial fields, frequency sparsity is computed via the 2D real FFT.

**Frequency Grid.** Let $\hat{\mathbf{u}} = \text{rfft2}(\mathbf{u})$ with shape $(H, W/2 + 1)$. The frequency coordinates are:

$$k_y \in \{0, 1, \ldots, H/2, -H/2+1, \ldots, -1\}, \tag{24}$$
$$k_x \in \{0, 1, \ldots, W/2\}. \tag{25}$$

The frequency radius is $\|\mathbf{k}\| = \sqrt{k_y^2 + k_x^2}$.

**Bandlimited Sparsity.** The in-band region $\mathcal{B} = \{\mathbf{k} : \|\mathbf{k}\| \leq k_{\max}\}$ and out-of-band $\bar{\mathcal{B}} = \{\mathbf{k} : \|\mathbf{k}\| > k_{\max}\}$:

$$\mathcal{R}_{\text{band}}(\mathbf{u}) = \frac{\sum_{\mathbf{k}\in\mathcal{B}} |\hat{u}_{\mathbf{k}}|}{\sqrt{\sum_{\mathbf{k}\in\mathcal{B}} |\hat{u}_{\mathbf{k}}|^2 + \epsilon}} + \beta_1 \cdot \frac{\sum_{\mathbf{k}\in\bar{\mathcal{B}}} |\hat{u}_{\mathbf{k}}|^2}{\sum_{\mathbf{k}} |\hat{u}_{\mathbf{k}}|^2 + \epsilon}. \tag{26}$$

**Frequency Exclusion.** For layer $\ell$, let $\{(\bar{k}_y^{(j)}, \bar{k}_x^{(j)})\}_{j=1}^J$ be frequencies discovered by previous layers. The exclusion region with radius $r_{\text{exc}}$ is:

$$\mathcal{E}^{(\ell)} = \bigcup_{j=1}^J \left\{ \mathbf{k} : \sqrt{(k_y - \bar{k}_y^{(j)})^2 + (k_x - \bar{k}_x^{(j)})^2} < r_{\text{exc}} \right\}. \tag{27}$$

Due to conjugate symmetry of real signals, both $(k_y, k_x)$ and $(-k_y, k_x)$ are excluded. The exclusion penalty uses weight $\beta_2$:

$$\mathcal{P}_{\mathcal{E}}^{(\ell)} = \beta_2 \cdot \frac{\sum_{\mathbf{k}\in\mathcal{E}^{(\ell)}} |\hat{u}_{\mathbf{k}}|^2}{\sum_{\mathbf{k}} |\hat{u}_{\mathbf{k}}|^2 + \epsilon}. \tag{28}$$

After training each HF peeling layer, dominant frequencies are identified by computing the 2D FFT of the mean HF output over the training set, excluding the DC component, and extracting the top-$k_\ell$ unique frequency locations accounting for conjugate symmetry. These are then added to the exclusion set for subsequent layers.

Table 3: Default hyperparameters for the SENDAI architecture.

| Component | Parameter | Value |
|---|---|---|
| LSTM Encoder | Hidden dimension $d_z$ | 32 |
| | Number of layers $K$ | 2 |
| | Dropout rate | 0.1 |
| | Temporal lags $L$ | 5 |
| LF Decoder | Hidden layers | [256, 256] |
| | Activation | ReLU |
| GAN | Generator hidden | 64 |
| | Discriminator hidden | 64 |
| INR (per HF layer) | Latent dimension $d_{\mathrm{HF}}$ | 64 |
| | Encoder hidden | [128, 128] |
| | Decoder hidden | [256, 256, 128] |
| | PE frequencies $L$ | 16 |
| | PE max frequency $\sigma_{\mathrm{max}}$ | 8.0 |
| | Scale $\gamma$ init | 0.1 |
| HF Training | Warmup epochs $E_{\mathrm{warm}}$ | 100 |
| | $\lambda_{\mathrm{sp}}$ | 0.05 |
| | $\lambda_{\mathrm{sm}}$ | 0.1 |
| | Fine-tune $\lambda'_{\mathrm{sp}}$ | 0.005 |
| Sparsity Penalties | Out-of-band $\beta_1$ | 100 |
| | Exclusion $\beta_2$ | 100 |
| | Top-k weight $\lambda_{\mathrm{topk}}$ | 10.0 |
| | Exclusion radius $r_{\mathrm{exc}}$ | 2.0 |
| Adaptive $k_\ell$ | Bandwidth tolerance $\Delta k$ | 2.0 |
| | Energy threshold $\rho$ | 0.8 |
| Optimization | Learning rate | $10^{-4}$ |
| | Batch size | 16 |
| | Optimizer | AdamW |

**Top-$k_\ell$ Sparsity.** For peeling layer $\ell$ with adaptively selected $k_\ell$ modes:

$$\mathcal{R}_{\mathrm{topk}}(\mathbf{u}; k_\ell) = 1 - \frac{\sum_{i=1}^{k_\ell} |\hat{u}_{(i)}|^2}{\sum_{\mathbf{k}} |\hat{u}_{\mathbf{k}}|^2 + \epsilon}, \tag{29}$$

where $|\hat{u}_{(i)}|$ is the $i$-th largest Fourier magnitude.

### F.5 SMOOTHNESS REGULARIZATION OPTIONS

Three smoothness regularization strategies are supported, with selection guided by prior knowledge of the expected HF field characteristics:

**Gradient (Total Variation).** For fields with sharp edges or discontinuities:

$$\mathcal{R}_{\mathrm{grad}}(\mathbf{u}) = \frac{1}{HW} \sum_{i,j} \left[ (u_{i,j+1} - u_{i,j})^2 + (u_{i+1,j} - u_{i,j})^2 \right]. \tag{30}$$

This penalizes gradients uniformly, promoting piecewise constant solutions.

**Laplacian (Curvature).** For fields expected to be smooth with gentle variations:

$$\mathcal{R}_{\mathrm{lap}}(\mathbf{u}) = \frac{1}{(H-2)(W-2)} \sum_{i,j} \left[ u_{i+1,j} + u_{i-1,j} + u_{i,j+1} + u_{i,j-1} - 4u_{i,j} \right]^2. \tag{31}$$

This penalizes curvature (second derivatives) rather than gradients, allowing linear ramps and sharp but smooth features while suppressing high-frequency oscillations.

Table 4: Computational cost and model complexity comparison.

| SENDAI Jr. | | SENDAI | |
|---|---|---|---|
| *Model Complexity* | | *Model Complexity* | |
| SHRED (LSTM + Decoder) | 1,149.0 K | SHRED (LSTM + Decoder) | 1,149.0 K |
| DA Transform | 4.2 K | DA Transform | 4.2 K |
| | | HF Peeling Layers | 334.5 K |
| **Total parameters** | **1,153.2 K** | **Total parameters** | **1,487.7 K** |
| *Training Time* | | *Training Time* | |
| Stage 1 (SHRED) | 2.76 sec | Stage 1 (SHRED) | 2.85 sec |
| Stage 2 (DA-SHRED + GAN) | 10.26 sec | Stage 2 (DA-SHRED + GAN) | 14.18 sec |
| | | Stage 3 (Hierarchical HFP + INR) | 25 min 50 sec |
| **Total** | **15.94 sec** | **Total** | **26 min 12 sec** |
| *Hardware* | | | |
| CPU: Apple M4, Memory: 24 GB | | | |

**Bilateral (Edge-Preserving).** For fields with both smooth regions and sharp boundaries:

$$\mathcal{R}_{\text{bilateral}}(\mathbf{u}) = \sum_{i,j} \left[ \text{Huber}_\delta(u_{i,j+1} - u_{i,j}) + \text{Huber}_\delta(u_{i+1,j} - u_{i,j}) \right], \tag{32}$$

where $\text{Huber}_\delta(x) = \frac{1}{2}x^2$ if $|x| < \delta$, else $\delta(|x| - \frac{\delta}{2})$. The threshold $\delta$ controls the transition between quadratic (small gradients) and linear (large gradients) penalization, preserving edges while smoothing homogeneous regions.

### F.6 HYPERPARAMETER CONFIGURATION

Table 3 summarizes the default hyperparameters.

### F.7 COMPUTATIONAL COST AND MODEL COMPLEXITY

Table 4 summarizes the computational cost and model complexity.

## G SYNTHETIC VALIDATION: EXTENDED DETAILS

This appendix provides comprehensive details for the synthetic validation experiments presented in Section 4.1.

### G.1 TRAVELING WAVE SYSTEM: DATA GENERATION

The synthetic traveling wave system is generated on a spatial domain $x \in [0, 2\pi]$ with $N = 128$ grid points and temporal domain $t \in [0, 10]$ with $\Delta t = 0.05$ (200 timesteps). The full three-mode field is:

$$u(x,t) = \sin(2x - t) + 0.4\sin(5x - 3t) + 0.25\sin(11x - 7t), \tag{33}$$

where the different temporal frequencies $\omega_1 = 1$, $\omega_2 = 3$, $\omega_3 = 7$ ensure the three modes remain distinguishable as the system evolves. The simulation model contains only the first term, representing a simplified physics model that misses intermediate and fine-scale dynamics.

Figure 6 compares two HF correction strategies: joint discovery versus hierarchical peeling. Both achieve comparable reconstruction accuracy in terms of RMSE, but the joint approach produces a relatively noisy frequency spectrum with energy spread across multiple modes beyond the targets, whereas hierarchical peeling yields clean outputs for each mode. This spectral purity translates to improved fine-scale reconstruction, particularly in regions where modes interfere constructively or destructively.

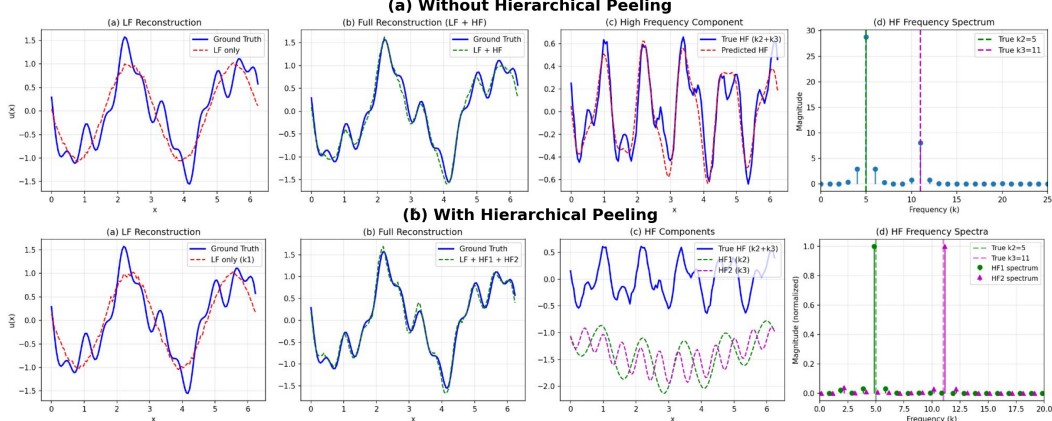

Figure 6: Comparison of joint and hierarchical frequency discovery on the three-mode traveling wave system for a single time-point reconstruction. **(a)** the frequency spectrum shows energy leakage to non-target modes. **(b)** modes are discovered sequentially, yielding spectrally clean outputs and improved fine-scale fidelity. In third panel, HF1 and HF2 show the individually learned components (unnormalized); their sum after scaling recovers the true HF.

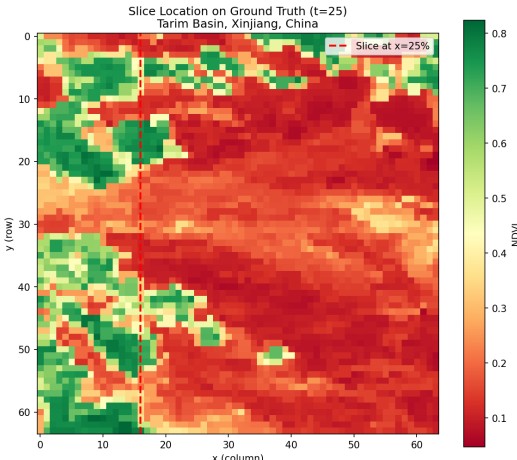

Figure 7: Slice location for hierarchical frequency peeling on a 1D NDVI slice from the Tarim Basin.

## G.2 NDVI SLICE EXTRACTION

For the NDVI slice experiment, we extract a 1D transect from the Tarim Basin site at $x = 25\%$ of the image width (column 16 of a $64 \times 64$ grid). This slice traverses a sharp mountain-basin boundary, capturing the heterogeneous vegetation structure characteristic of the landscape.

Both simulation data from the April–June period and ground truth from July–October are aligned to $T = 72$ timesteps for training. The HF residual exhibits:

- Range: $[-0.46, 0.86]$ NDVI units

- Standard deviation: 0.212

- Sharp discontinuities at mountain boundaries (pixels 10–20, 45–55)

- Temporal variability from phenology

G.3 Sensor Configuration

Both experiments employ an extremely sparse configuration of only $p = 3$ sensors. For the traveling wave system, this provides approximately 2.3% spatial coverage of the 128-point grid. For the NDVI slice, sensors are randomly placed as well. Despite this severe undersampling, both experiments successfully recover the missing frequency content, demonstrating the framework's ability to exploit temporal coherence for frequency recovery.

The LSTM encoder processes temporal histories of length $L = 20$ lags (traveling wave) or $L = 10$ lags (NDVI slice), exploiting temporal coherence to compensate for spatial sparsity.

For the NDVI experiment, INR decoders with Fourier positional encoding are used to produce spatially coherent outputs despite the sharp discontinuities in the target field.

G.4 Joint vs. Hierarchical Frequency Discovery

We compare two strategies for HF correction on the traveling wave system:

**Joint Discovery.** A single HF pathway with bandlimited sparsity discovers both $k_2$ and $k_3$ simultaneously. While achieving 84.7% RMSE improvement, the frequency spectrum shows energy spread across modes $k \in \{3, 4, 5, 10, 11, 12\}$ rather than concentrated solely at the targets. The combined HF output represents an entangled mixture with energy leakage.

**Sequential Peeling.** The hierarchical approach achieves 85.1% RMSE improvement with clean spectrum: $HF_1$ captures $k = 5$ with $> 95\%$ of its energy at the target mode, and $HF_2$ captures $k = 11$ similarly. This spectral purity enables physical interpretation of individual frequency contributions, downstream analysis of specific spectral components, and modular addition of peeling layers without retraining.

G.5 NDVI Transect Decomposition

The 1D NDVI slice from the Tarim Basin site illustrates how hierarchical frequency peeling decomposes a complex, noisy simulation–observation residual into interpretable components across scales.

**$HF_1$: Climate-Driven Seasonal Dynamics.** The first peeling layer captures the dominant residual variability, combining coherent temporal fluctuations with a broad spatial gradient along the transect. Its synchronized, episode-like oscillations are consistent with basin-wide meteorological forcing (e.g., temperature anomalies and intermittent precipitation events) acting across the slice, while the gradual late-season attenuation suggests a phenological convergence between spring-calibrated simulations and summer–autumn observations. The spatial amplitude structure further indicates modulation by the elevation gradient, supporting an interpretation of temperature-mediated phenological offsets that vary systematically across landscape zones.

**$HF_2$: Hydrological Persistence.** The second peeling layer is characterized by stronger, more persistent spatial stratification with relatively minimal temporal evolution, indicating a quasi-stationary control on the residual. In the hyperarid Tarim Basin, such stable structure could be plausibly explained by differential water availability governed by proximity to snow-melt, groundwater access, and topographic accumulation of runoff.

**$HF_3$: Edaphic and Microsite Heterogeneity.** The third peeling layer isolates finer-scale, temporally invariant spatial heterogeneity, pointing to localized controls that persist over the season. We attribute this component to edaphic and microsite factors such as soil texture, salinity, nutrient level, and geomorphic microhabitats (e.g., abandoned channels or alluvial features) that influence vegetation independently of broad climate and hydrological gradients. Predominantly negative anomalies are consistent with systematic overestimation by the low-frequency reconstruction in sub-pixel heterogeneous patches that cannot be resolved without an explicit multiscale residual model.

# H   SENDAI JR. SITE-SPECIFIC RESULTS

This appendix provides detailed analyses and qualitative reconstruction results for the two sites evaluated using the SENDAI Jr. pipeline. Performance is assessed using both RMSE and the Structural Similarity Index Measure (SSIM), with SSIM serving as the primary indicator of reconstruction quality. As explained in the main text, baseline methods fail to preserve the topological structure of spatial patterns despite achieving moderate RMSE values. IDW-based methods exhibit "bullseye" artifacts centered at sensor locations. Kriging produces overly smooth reconstructions that obscure sharp boundaries. SSIM captures the preservation of spatial patterns, textures, and structural information that RMSE alone cannot assess—critically important for remote sensing applications where topological fidelity determines downstream analysis utility.

## H.1   CENTRAL VALLEY, CALIFORNIA, USA

This irrigated cropland site achieves RMSE of 0.1068 and SSIM of 0.5747, representing a 120% SSIM improvement over the best-performing baseline (SG+IDW at 0.2612). This substantial structural improvement indicates that SENDAI Jr. successfully preserves field boundaries, irrigation patterns, and vegetation gradients that baseline methods systematically destroy.

Figure 11 shows baseline performance on the Central Valley site. While this site exhibits less extreme heterogeneity than the Tarim Basin, the baseline methods still produce characteristic artifacts: IDW-based methods show bullseye patterns around sensor locations (SSIM: 0.2612 and 0.2504), while Kriging over-smooths field boundaries resulting in the lowest SSIM (0.0922). The poor SSIM values of baseline methods—despite moderate RMSE—demonstrate the critical importance of structural similarity metrics for evaluating reconstruction quality.

Figure 12 presents qualitative reconstruction results across four equally-spaced temporal frames within the ground truth observation period. The reconstructed fields preserve the essential spatial heterogeneity of the ground truth, including field boundaries and vegetation gradients, despite access to only 64 point measurements per frame. The high SSIM value (0.5747) quantitatively confirms this visual assessment of structural preservation.

## H.2   GUADALQUIVIR VALLEY, SPAIN

This Mediterranean site, characterized by reversed phenological timing relative to temperate Northern Hemisphere regions, achieves RMSE of 0.1474 and SSIM of 0.3655. The SSIM represents a 98% improvement over the best baseline (SG+IDW at 0.1849). The framework successfully adapts from winter-spring simulation conditions (February–April) to autumn ground truth observations (September–December), demonstrating robustness to non-standard seasonal calendars.

Figure 13 shows baseline performance on this site. The mixture of irrigated agriculture and natural vegetation creates heterogeneous patterns that baseline methods fail to faithfully reproduce. All baseline methods achieve SSIM below 0.19, indicating severe structural degradation. Kriging again shows the largest disconnect between RMSE and SSIM, achieving moderate RMSE (0.1481) but poor SSIM (0.0878).

Figure 14 demonstrates SENDAI Jr. reconstruction for the Guadalquivir Valley, where the reversed Mediterranean phenological calendar requires adaptation from winter-spring greenness to autumn ground truth conditions. The model recovers coherent spatial patterns including agricultural field boundaries and regional vegetation gradients, as reflected in the substantially improved SSIM.

## H.3   SUMMARY OF SENDAI JR. PERFORMANCE

Across both sites, SENDAI Jr. demonstrates consistent superiority over baseline methods in both RMSE and SSIM metrics, with particularly pronounced advantages in structural preservation:

- **Central Valley**: 120% SSIM improvement, preserving irrigated field boundaries
- **Guadalquivir Valley**: 98% SSIM improvement, capturing Mediterranean agricultural structure

The large gap between baseline RMSE and SSIM performance—particularly for Kriging—demonstrates that traditional error metrics inadequately capture reconstruction quality for heterogeneous landscapes. SENDAI Jr.'s success in improving both metrics simultaneously indicates that the framework learns physically meaningful spatial priors rather than simply minimizing point-wise error.

## I  SENDAI SITE-SPECIFIC RESULTS

This appendix provides detailed analyses and qualitative reconstruction results for the two sites evaluated using the full SENDAI hierarchical multiscale DA-SHRED architecture with INR. These sites exhibit complex phenological dynamics, sub-seasonal variability, or pronounced spatial heterogeneity that require high-frequency correction beyond what SENDAI Jr. can provide. SSIM serves as our primary performance indicator, capturing the preservation of sharp boundaries, fine-scale features, and topological structure.

### I.1  IMPERIAL VALLEY, CALIFORNIA, USA

This dry and hot irrigated agriculture site exhibits consistent improvement through the hierarchical pipeline. SENDAI Jr. achieves SSIM of $0.4041 \pm 0.0248$, which increases to $0.4411 \pm 0.0499$ with HF peeling (+9.2%) and further to $0.4668 \pm 0.0390$ with the full SENDAI pipeline (+15.5% total improvement from SENDAI Jr.). RMSE improves correspondingly from $0.1708 \pm 0.0029$ to $0.1486 \pm 0.0063$.

The baseline methods achieve uniformly poor SSIM values: SG+IDW ($0.1123 \pm 0.0146$), HANTS+IDW ($0.1049 \pm 0.0145$), and Kriging ($0.0916 \pm 0.0190$). This represents a $4\times$ to $5\times$ improvement in structural similarity for the full SENDAI pipeline over baselines, demonstrating the framework's ability to preserve the rectilinear irrigation infrastructure that defines this landscape.

Figure 15 presents baseline reconstruction results for Imperial Valley. The contrast between irrigated fields and surrounding desert creates sharp NDVI boundaries that are lost in baseline reconstructions, manifesting as circular bullseye artifacts (IDW methods) or smooth gradients (Kriging). The low SSIM values quantify this structural failure.

Figure 16 shows the full hierarchical reconstruction pipeline. The HF pathway captures fine-scale field boundaries characteristic of the rectilinear irrigation infrastructure, with the learned HF component exhibiting coherent spatial structure aligned with field edges rather than random noise.

### I.2  TARIM BASIN, CHINA

This continental site presents the most pronounced contrast between SENDAI Jr. and SENDAI performance. SENDAI Jr. achieves SSIM of $0.3505 \pm 0.0269$, which increases to $0.4257 \pm 0.0159$ with HF peeling (+21.5%) and to $0.4777 \pm 0.0205$ with the full pipeline (+36.3% total improvement). This is the largest SSIM improvement across all sites, reflecting the critical importance of hierarchical high-frequency correction for landscapes with sharp boundaries.

The hierarchical pipeline also achieves substantial RMSE improvement: from $0.1827 \pm 0.0046$ (SENDAI Jr.) to $0.1208 \pm 0.0109$ (SENDAI), a 33.9% reduction. This demonstrates that the high-frequency corrections are not merely cosmetic but represent genuine improvements in reconstruction accuracy.

Baseline methods fail catastrophically on this site in terms of structural preservation: Kriging achieves the worst SSIM (0.0449), producing nearly featureless smooth fields despite moderate RMSE. The sharp mountain-basin boundaries that define this landscape are completely unresolved by all baseline approaches.

Figure 17 illustrates the failure of baseline methods on this challenging site. The oasis-desert transition creates extreme spatial gradients that interpolation-based methods cannot capture. IDW methods produce pronounced bullseye artifacts, while Kriging eliminates all boundary information.

Figure 18 presents the full hierarchical reconstruction pipeline. The HF component captures the sharp oasis-desert boundaries that the smooth LF decoder fails to resolve. The learned corrections exhibit

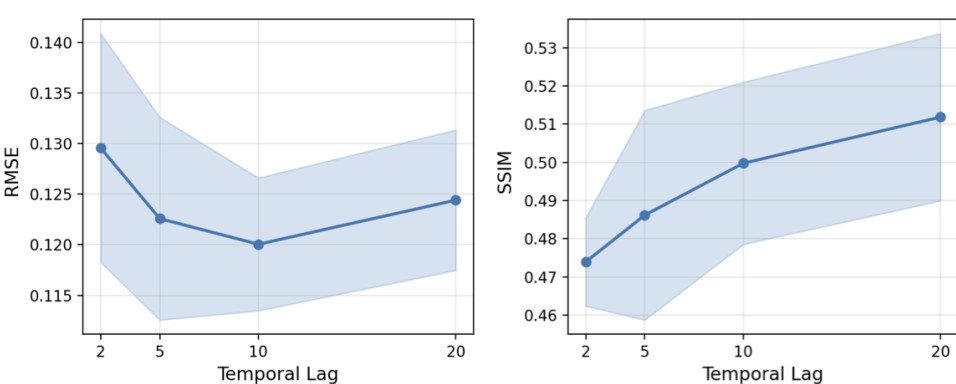

Figure 8: Effect of temporal lag $L$ on full SENDAI reconstruction (Tarim Basin). Shaded regions represent $\pm 1$ std.

spatially coherent structure aligned with the landscape's topological features, confirming that the peeling layers discover physically meaningful high-frequency content rather than noise.

### I.3   SUMMARY OF SENDAI PERFORMANCE

The hierarchical SENDAI framework demonstrates consistent improvements over both baselines and SENDAI Jr. across both challenging sites:

- **Imperial Valley**: 15.5% SSIM improvement from SENDAI Jr., $4.2\times$ improvement over best baseline
- **Tarim Basin**: 36.3% SSIM improvement from SENDAI Jr., $3.7\times$ improvement over best baseline

## J   HYPERPARAMETER SENSITIVITY AND NOISE ROBUSTNESS

We conducted a series of ablation studies on the Tarim Basin site to examine the sensitivity of the full SENDAI pipeline to sensor count, temporal lag and maximum target frequency on the HF pathway, as well as robustness to sensor noise. All reported metrics are computed on the held-out ground-truth evaluation set and averaged over three independent runs, except for the sensor-number sensitivity analysis, which uses five runs.

**Sensor number.** Figure 4a sweeps $n \in \{8, 16, 32, 64, 128, 256\}$, showing SSIM performance as a function of sensor count. Both the full dataset and validation set exhibit a clear positive trend: reconstruction quality improves with increasing sensor density. For our main experiments, we selected 64 sensors as a conservative configuration. At this density, the model achieves mean SSIM values of approximately 0.53 (full) and 0.49 (validation), representing a reasonable trade-off between reconstruction accuracy and practical deployment constraints. While higher sensor counts yield improved performance, the marginal gains must be weighed against increased instrumentation costs and maintenance requirements in field applications.

**Temporal lag.** Figure 8 sweeps $L \in \{2, 5, 10, 20\}$. SSIM improves monotonically with increasing lag, consistent with Takens' embedding theorem: longer sensor histories provide a more faithful embedding of the underlying dynamics. RMSE shows a similar improving trend up to $L=10$. We adopt $L=5$ as the default throughout the paper as a conservative choice.

**Maximum target frequency.** Figure 9 sweeps $k_{\max} \in \{4, 8, 12, 16, 20, 24\}$. Both RMSE and SSIM remain relatively stable across the range, with slight degradation at the highest $k_{\max}$ as the expanded band admits modes that the sparse sensor coverage cannot reliably constrain. The overall insensitivity

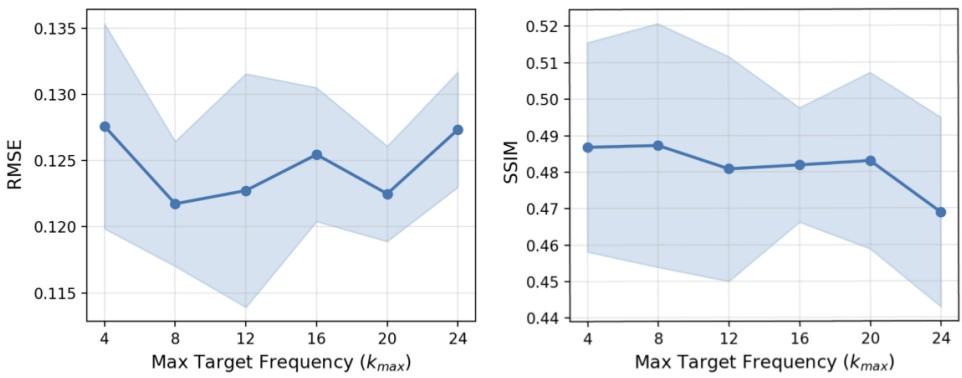

Figure 9: Effect of maximum target frequency $k_{\max}$ on full SENDAI reconstruction (Tarim Basin). Shaded regions represent $\pm 1$ std.

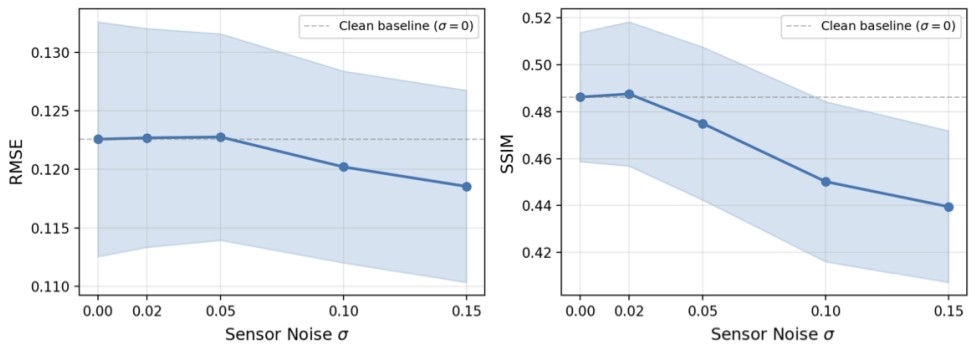

Figure 10: Sensor noise robustness of SENDAI (Tarim Basin). The model is trained on clean data and evaluated with Gaussian noise of varying $\sigma$ added to sensor observations. Dashed line indicates the clean baseline. Shaded regions indicate $\pm 1$ standard deviation over three independent runs.

implies that SENDAI's spectral sparsity regularization and frequency exclusion zones are preventing the model from overfitting.

**Sensor noise robustness.** Figure 10 trains on clean data and evaluates on sensors corrupted with additive Gaussian noise $\sigma \in \{0, 0.02, 0.05, 0.10, 0.15\}$ in raw NDVI units, thereby emulating deployment under degraded sensor quality. Both RMSE and SSIM remain stable across the noise levels tested. The robustness is expected given that the MODIS data already contains inherent sensor noise and atmospheric contamination, and the SHRED temporal unit naturally attenuates uncorrelated noise over the lag window. These results confirm that the hierarchical corrections are not artifacts of overfitting to sensor noises.

## K   HARDWARE EFFICIENCY: EXTENDED DISCUSSION

This appendix provides additional operational scenarios for the hardware efficiency paradigm introduced in Section 5.2. Consider a satellite acquiring a $64 \times 64$ pixel scene (4,096 values). Under our sparse sensing protocol, only 64 sensor measurements need be transmitted—a $64\times$ reduction before any conventional compression. For deep-space exploration missions where communication bandwidth constrains scientific return (De Cola et al., 2011; Xie et al., 2021), SENDAI-like architec-

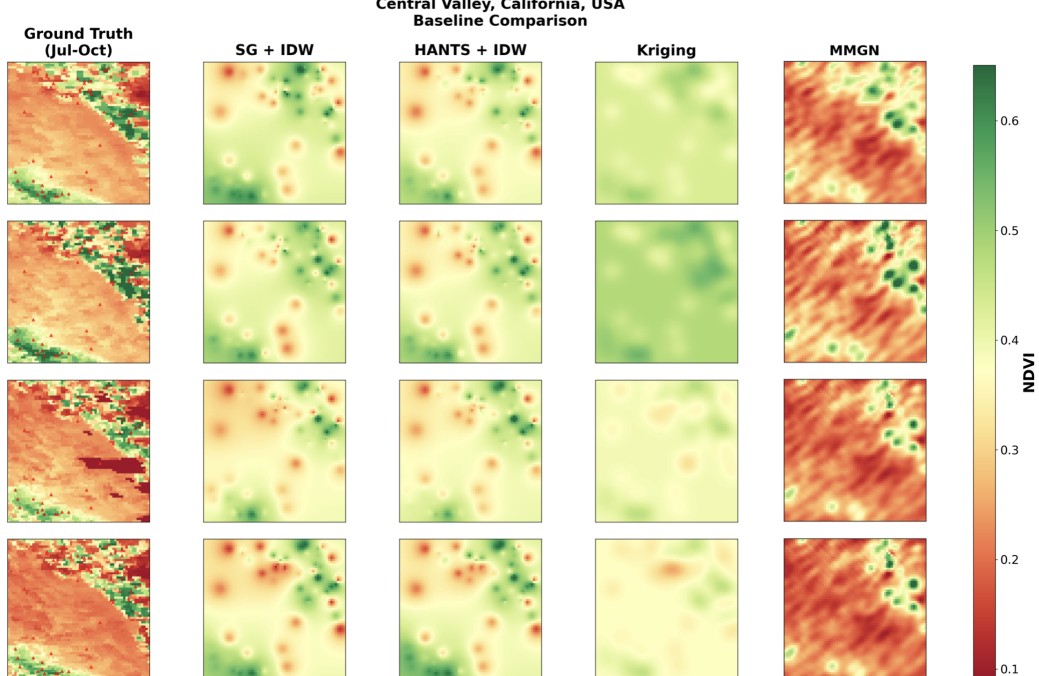

Figure 11: Baseline reconstruction comparison for Central Valley, California. Each row represents an equally-spaced temporal frame within the ground truth period. The irrigated cropland landscape exhibits field-scale heterogeneity that baseline interpolation methods fail to faithfully reproduce. Note the bullseye artifacts in IDW-based methods and over-smoothing in Kriging, which are reflected in their low SSIM values.

tures could enable substantial compression while preserving spatial structure essential for scientific interpretation.

## L  IMPLICIT NEURAL REPRESENTATIONS FOR GEOSPATIAL DATA

The coordinate-based implicit neural representation (INR) employed in the high-frequency pathway represents an emerging paradigm for geospatial data with substantial unexplored potential (Sitzmann et al., 2020; Tancik et al., 2020). Unlike discrete gridded representations, INRs parameterize spatial fields as continuous functions, enabling queries at arbitrary coordinates and natural handling of irregular geometries.

Recent work has demonstrated INR applicability across Earth science domains: potential field geophysics (Smith et al., 2025), species distribution modeling (Cole et al., 2023), and climate data compression (Mostajeran et al., 2025). Our application to high-frequency correction in satellite imagery suggests additional utility: INRs can learn spatially coherent patterns from sparse sensor residuals, producing smooth interpolation without the localized artifacts characteristic of direct MLP regression.

The combination of INR decoders with recurrent encoders of sensor time-histories represents a hybrid architecture with broader applicability. The recurrent encoder captures temporal dynamics (phenological trends, seasonal patterns); the INR decoder produces spatially coherent instantaneous fields. This separation of temporal and spatial processing may prove advantageous across spatiotemporal reconstruction problems where temporal and spatial structure exhibit distinct characteristics.

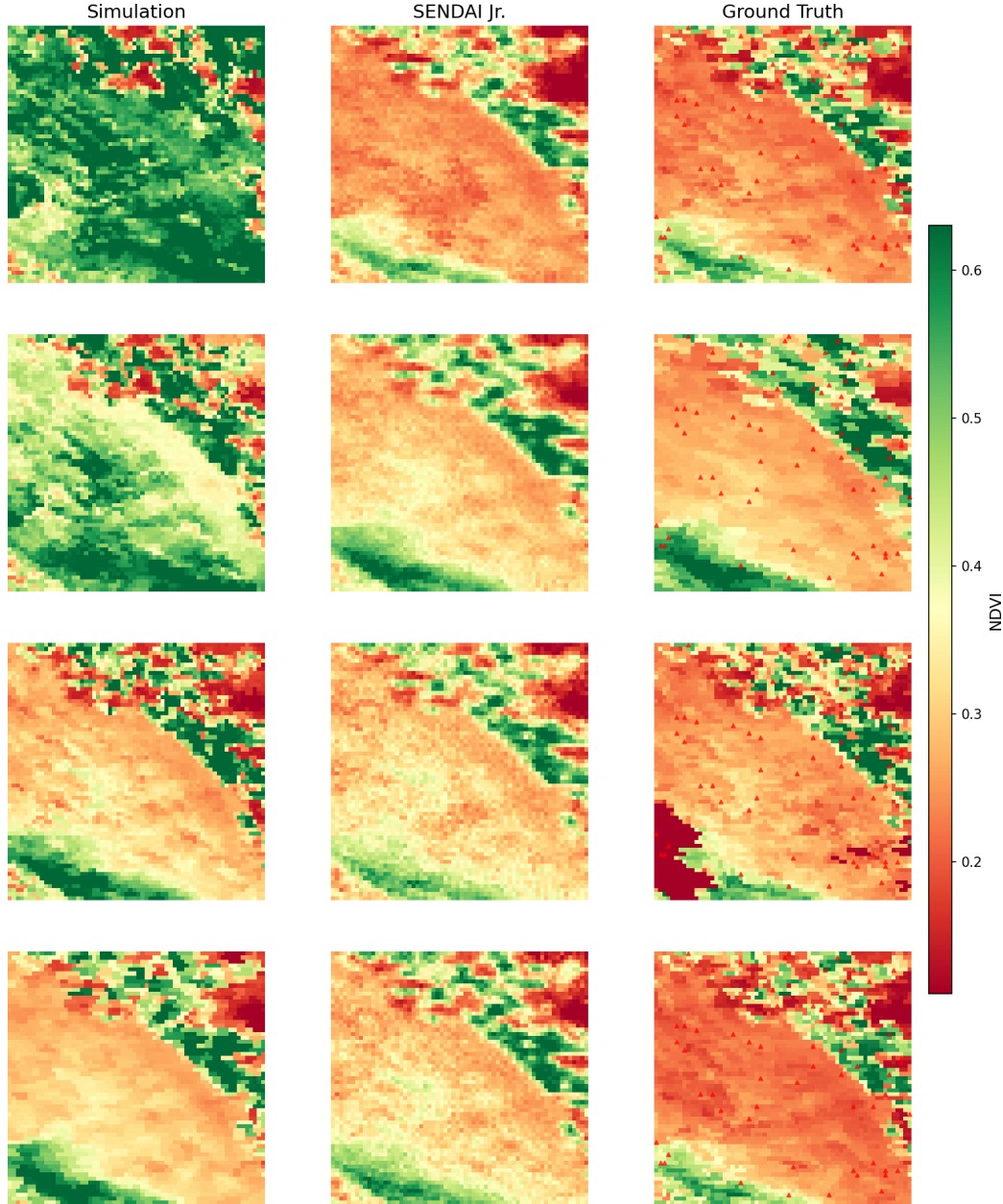

Figure 12: SENDAI Jr. reconstruction for Central Valley, California. Each row represents an equally-spaced temporal frame within the ground truth period (July–October). Columns show simulation-period reference (April–June), SENDAI Jr. reconstruction, and ground truth. Red markers indicate sensor locations. The SSIM of 0.5747 reflects the preservation of field-scale spatial structure.

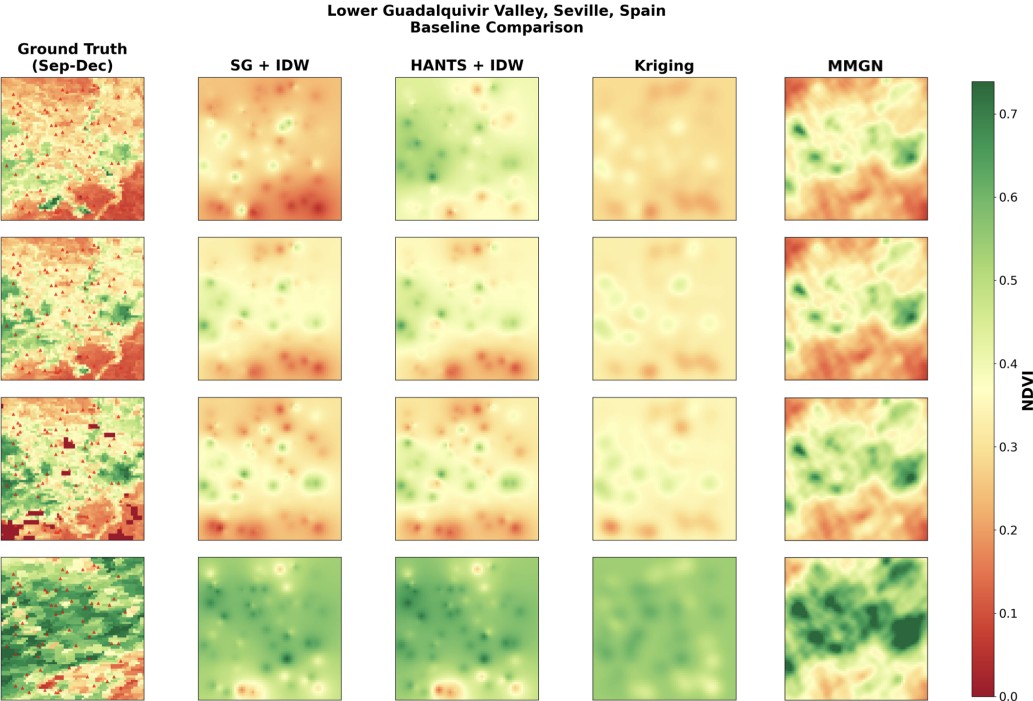

Figure 13: Baseline reconstruction comparison for the Guadalquivir Valley, Spain. Each row represents an equally-spaced temporal frame. The mixed agricultural landscape with varying field sizes challenges interpolation-based methods. All baselines achieve SSIM below 0.19, indicating severe structural degradation.

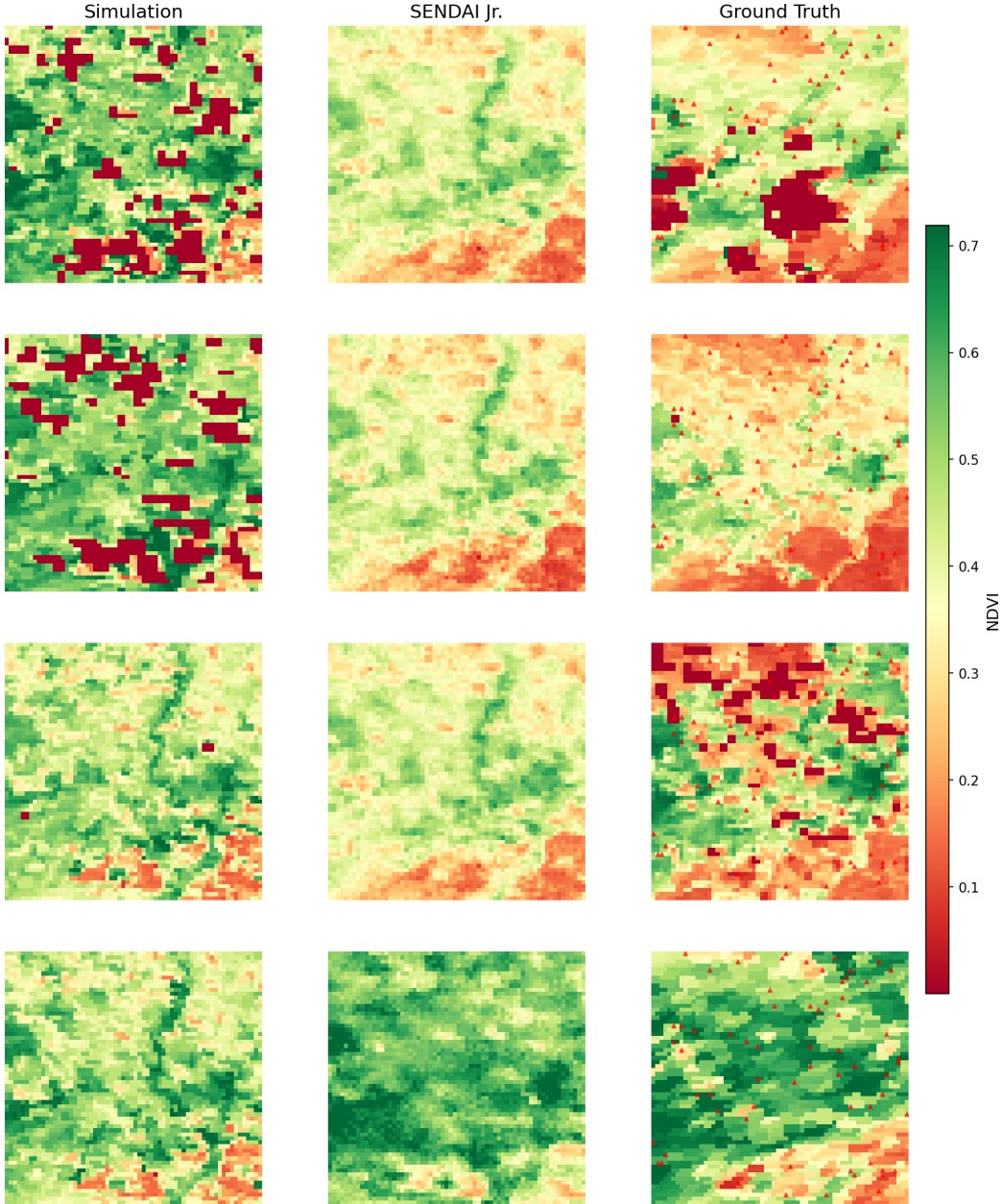

Figure 14: SENDAI Jr. reconstruction for the Guadalquivir Valley, Spain. Temporal frames span September–December ground truth observations, reconstructed from February–April simulation training. The model recovers coherent spatial patterns including agricultural field boundaries and regional vegetation gradients, achieving SSIM of 0.3655.

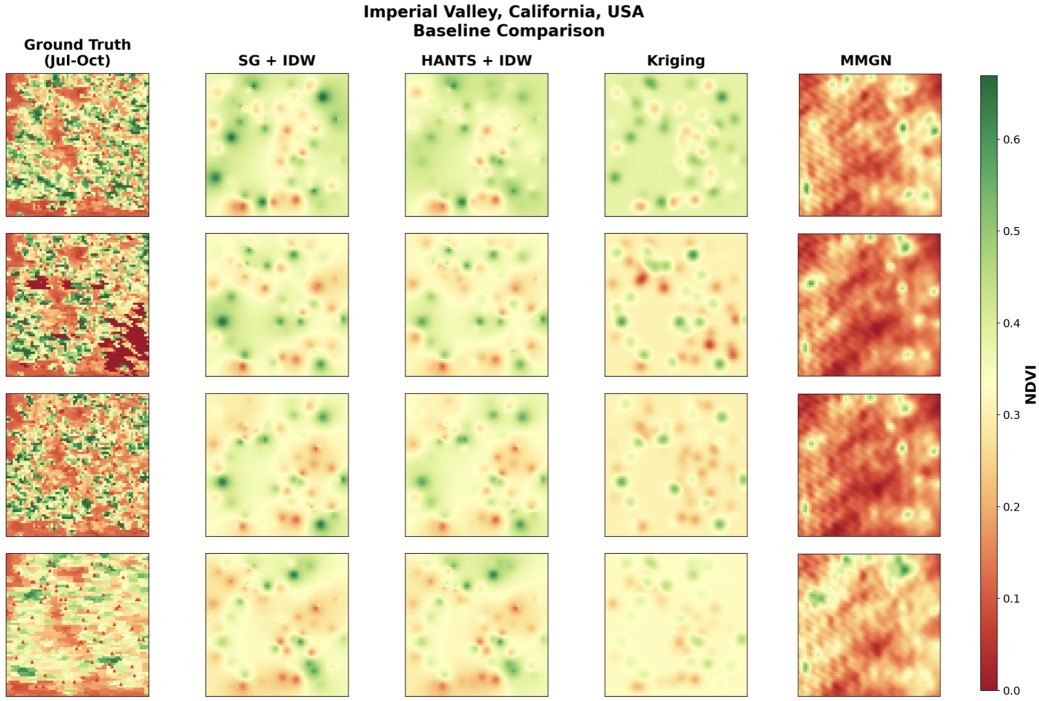

Figure 15: Baseline reconstruction comparison for Imperial Valley, California. Each row corresponds to an equally-spaced temporal frame. The rectilinear irrigation infrastructure creates sharp NDVI boundaries that baseline methods fail to preserve, producing either circular artifacts or smooth gradients. Baseline SSIM values range from 0.0916 to 0.1123.

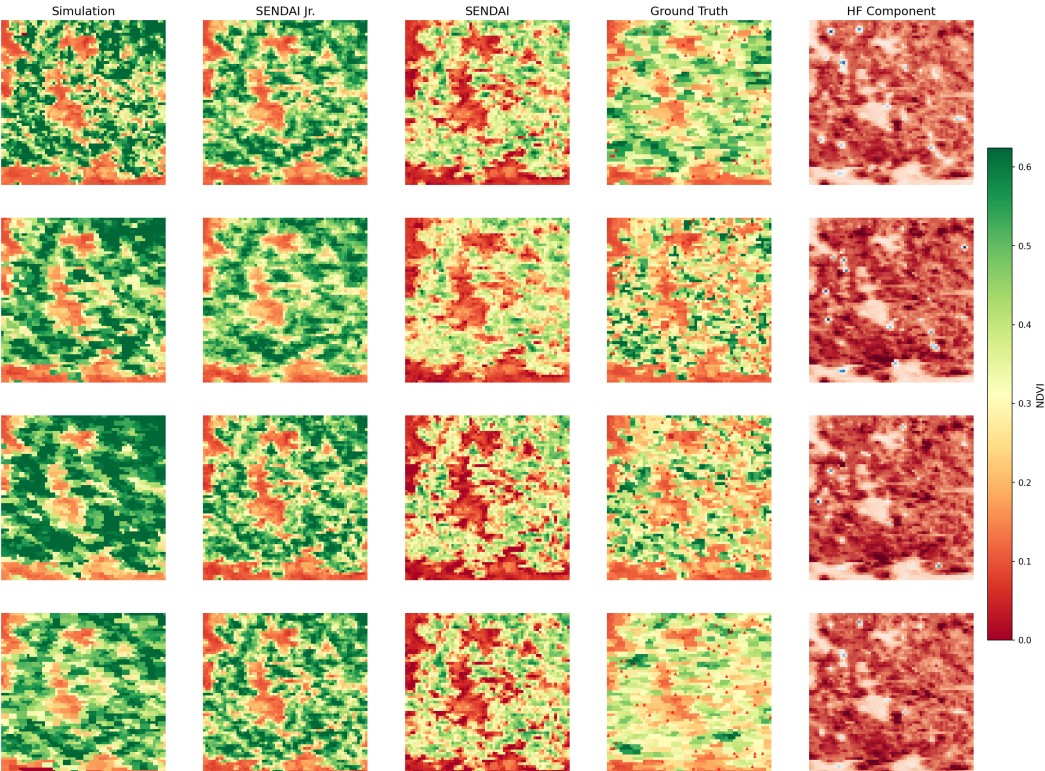

Figure 16: Full SENDAI hierarchical multiscale DA-SHRED reconstruction for Imperial Valley, California. Each row corresponds to an equally-spaced temporal frame during July–October. Columns show: simulation reference (April–June), SENDAI Jr. stage, full SENDAI hierarchical reconstruction, ground truth, and the learned HF correction component. SENDAI captures fine-scale field boundaries, achieving SSIM of 0.4668 in the full pipeline.

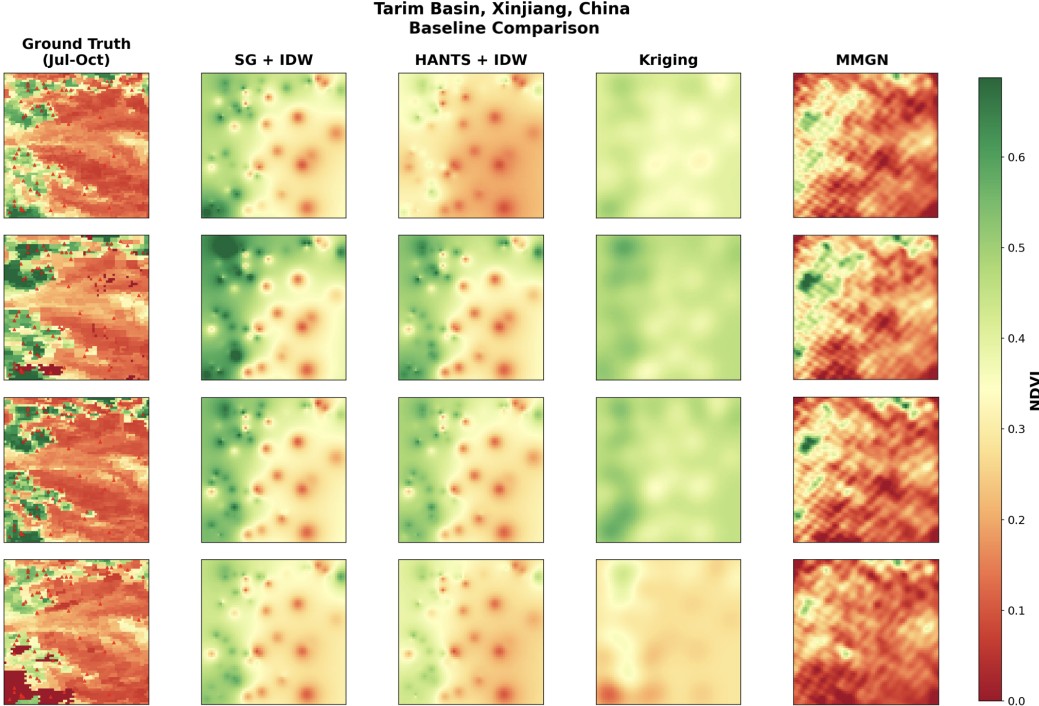

Figure 17: Baseline reconstruction comparison for the Tarim Basin, Xinjiang, China. Each row represents an equally-spaced temporal frame. The sharp mountain-basin boundaries that define this landscape are completely lost in all baseline reconstructions, demonstrating the fundamental limitations of interpolation-based approaches for heterogeneous terrain. Kriging achieves SSIM of only 0.0449.

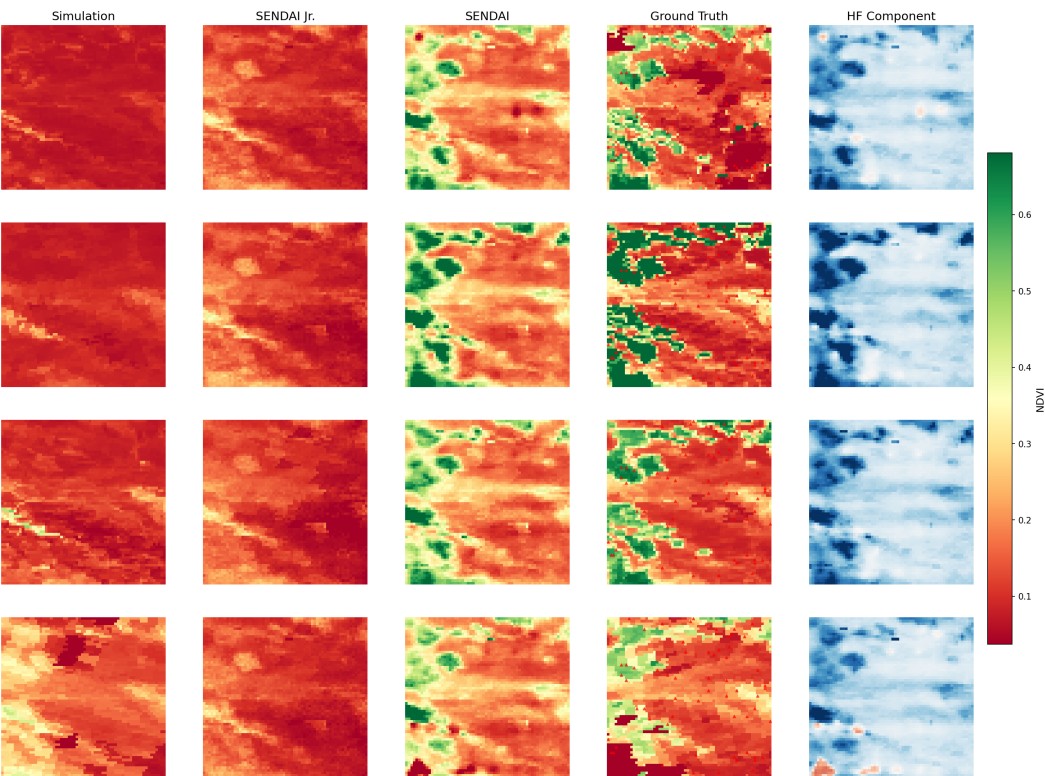

Figure 18: Full SENDAI hierarchical multiscale DA-SHRED reconstruction for the Tarim Basin, Xinjiang, China. Temporal frames span July–October ground truth observations. The HF component captures sharp oasis-desert boundaries that the smooth LF decoder fails to resolve, with the full pipeline achieving SSIM of 0.4777 and 33.9% RMSE improvement over SENDAI Jr.

