# OpenReview forum: "SENDAI: A Hierarchical Sparse-measurement, EfficieNt Data AssImilation Framework"
_ICLR.cc/2026/Workshop/FM4Science — ICLR 2026 Workshop FM4Science Poster_

### Official Review · Reviewer_zUwy · 2026-02-22

**Rating:** 6
**Confidence:** 4

**Review:**

Pros:
- The paper tackles an important and practically relevant problem of reconstructing spatiotemporal fields under extreme sparse sensing, which is highly relevant to Earth observation and scientific machine learning applications.
- The motivation is clearly articulated and grounded in real operational constraints such as sparse observations, domain shift, and limited computational resources, which strengthens the practical relevance of the work.
- The paper is generally well structured, with clear figures and mathematical formulations that make the methodology understandable despite its complexity.


Cons:
- The novelty relative to prior work in data assimilation, implicit neural representations, and multiscale modeling appears somewhat incremental because the framework largely combines existing components rather than introducing fundamentally new modeling principles.
- The paper lacks theoretical analysis such as convergence guarantees, identifiability discussion, or reconstruction error bounds, which limits the depth of the technical contribution.
- The experimental evaluation does not include strong modern baselines such as neural operators, diffusion-based reconstruction models, or transformer-based spatiotemporal architectures, making it difficult to assess state-of-the-art performance.
- The ablation studies are limited, which makes it unclear how much each component (e.g., GAN alignment, hierarchical peeling, spectral regularization) contributes to the overall performance gains.
- The claims of generalization and transferability are somewhat overstated given that experiments are conducted on a small number of sites and only a single variable (NDVI).
- The framework does not provide uncertainty quantification, which is an important aspect for data assimilation and scientific decision-making applications.
- Some methodological details such as hyperparameter sensitivity, training stability, and potential failure cases are not sufficiently discussed in the main text.
- The computational efficiency claims would be stronger with direct runtime comparisons against neural baselines rather than qualitative statements.

---

### Official Review · Reviewer_2p2F · 2026-02-23
**Nice high-frequency reconstruction**

**Rating:** 5
**Confidence:** 5

**Review:**

The paper details a lightweight data assimilation framework designed to reconstruct full spatiotemporal fields from extremely sparse sensor measurements.

Strengths

It’s nice to see the model successfully reconstructs complex fields from a mere 64 sensor data points. It’s also nice to see the shallow architecture bypasses the need for GPU clusters and trains in minutes on standard CPU hardware.

Structural Fidelity: The high-frequency peeling successfully preserves sharp topological boundaries and massively improves SSIM metrics.

Weaknesses

The architecture strings together an LSTM, a GAN, sequential frequency peeling, and an INR decoder. This multi-stage, disjointed approach feels unnecessarily complex for a model with only 1.4 million parameters.

Relying on LSTMs and small GANs is counterintuitive for modern foundation models. While the authors defend this based on computational efficiency, they fail to explore whether a unified, modern lightweight architecture could solve the problem more elegantly without disjointed pathways.

Traditional interpolation baselines like Kriging and IDW inherently smooth over sharp boundaries. Because SENDAI is explicitly designed to preserve high-frequency edges, testing against these older methods guarantees an artificial win on the SSIM metric.

The only modern deep learning baseline included is MMGN. However, the original MMGN was designed to operate with 5% to 50% sensor coverage. Testing it at 1.56% coverage effectively starves the baseline and weakens the comparative claim.

The empirical validation is strictly limited to NDVI data. The universality of the framework remains unproven for other crucial Earth observation variables like soil moisture or surface temperature.

Stationarity Assumption: The latent-space alignment assumes the underlying landscape topology remains relatively stationary across the simulation-to-reality domain shift. The model lacks a mechanism to handle sudden topological disturbances.

---

### Official Review · Reviewer_Dyko · 2026-02-23
**A Practical Framework for Sparse Spatiotemporal Field Reconstruction via Latent Alignment and Frequency Peeling**

**Rating:** 8
**Confidence:** 3

**Review:**

Summary

This paper proposes SENDAI, a hierarchical data assimilation framework for reconstructing spatiotemporal NDVI fields from extremely sparse observations. The approach decomposes reconstruction into (i) a low-frequency (LF) pathway based on a recurrent decoder trained on a source domain and adapted to a target domain via latent-space GAN alignment, and (ii) a high-frequency (HF) refinement stage that performs hierarchical “frequency peeling.” In the HF stage, the model iteratively fits residuals using coordinate-based implicit neural representations (INRs), enforcing frequency exclusion to encourage multiscale structure. Experiments on several MODIS NDVI regions show improved SSIM compared to classical interpolation and selected deep baselines under sparse sensor regimes.

Strengths

1. Clear architectural intuition. The LF/HF decomposition is conceptually clean and aligns with multiscale reconstruction ideas common in scientific modeling. The hierarchical peeling mechanism provides an interpretable framework for progressively refining spatial detail.

2. Interpretable inductive bias. The hierarchical frequency peeling and exclusion mechanism introduces a structured spectral prior rather than relying purely on black-box neural refinement.

3. Efficiency and practicality. The proposed system is relatively lightweight (small MLP INRs, shallow GAN alignment) and emphasizes operational feasibility, which is valuable for scientific applications with limited compute.

4. Interesting use of coordinate-based INRs. Using small coordinate networks as residual decoders is a reasonable design choice under sparse supervision, and the sequential residual fitting is technically coherent.

5. Empirical gains in structural metrics. The reported improvements in SSIM suggest the approach can produce visually coherent reconstructions even with very sparse sensors.

Major Concerns

1. Ill-posed inverse problem and lack of uncertainty analysis.
The reconstruction setting is extremely underdetermined (few sensors vs. dense field). While SENDAI introduces strong inductive biases (LF dynamics, spectral sparsity, INR smoothness), the paper does not sufficiently discuss non-uniqueness of solutions or quantify uncertainty. High-frequency layers may reflect model priors rather than physically verified structure.

2. “Simulation-to-real” framing is potentially misleading.
The source/target split appears closer to seasonal domain shift rather than true sim-to-real transfer involving physical simulators. Reframing this distinction would improve clarity and avoid overstating novelty.

3. Limited baseline coverage.
Comparisons focus mainly on classical interpolation and a limited set of learned models. Comparisons to more modern spatiotemporal or neural operator-style baselines would strengthen the empirical claims.

4. Hierarchical peeling novelty is somewhat incremental.
The HF stage can be interpreted as sequential residual regression (akin to matching pursuit or boosting) combined with coordinate INRs. While effective, the conceptual novelty relative to existing residual or multiscale decomposition approaches is not fully clarified.

---

### Decision · Program_Chairs · 2026-03-02

Accept (Poster)